biomathematics/health and disease
and epidemiology

multi-host, complexity, resilience, thresholds,
biting heterogeneity, spatial heterogeneity

**Author for correspondence:**
Edwin Michael
e-mail: emichael@nd.edu, emichael443@usf.edu

# Complexity and critical thresholds in the dynamics of visceral leishmaniasis

Shakir Bilal[1,2], Rocio Caja Rivera[2,3], Anuj Mubayi[4,5,6] and Edwin Michael[2,3]

[1]Amity Institute of Integrative Sciences and Health, Amity University Haryana, Gurugram (Manesar), Haryana 122 413, India
[2]Department of Biological Sciences, University of Notre Dame, Notre Dame, IN 46556, USA
[3]Center for Global Health Infectious Disease Research, University of South Florida, 3720 Spectrum Blvd, Suite 304, Tampa, FL 33612, USA
[4]College of Health Solutions, Arizona State University, Tempe, AZ 85281, USA
[5]Department of Mathematics, Illinois State University, IL, Normal, USA
[6]PRECISIONheor, Los Angeles, CA, USA

SB, 0000-0003-4425-0805

We study a general multi-host model of visceral leishmaniasis including both humans and animals, and where host and vector characteristics are captured via host competence along with vector biting preference. Additionally, the model accounts for spatial heterogeneity in human population and heterogeneity in biting behaviour of sandflies. We then use parameters for visceral leishmaniasis in the Indian subcontinent as an example and demonstrate that the model exhibits backward bifurcation, i.e. it has a human infection and a sandfly population threshold, characterized by a bi-stable region. These thresholds shift as a function of host competence, host population size, vector feeding preference, spatial heterogeneity, biting heterogeneity and control efforts. In particular, if control is applied through human treatment a new and lower human infection threshold is created, making elimination difficult to achieve, before eventually the human infection threshold no longer exists, making it impossible to control the disease by only reducing the infection levels below a certain threshold. A better strategy would be to reduce the human infection below a certain threshold potentially by early diagnosis, control animal population levels and keep the vector population under check. Spatial heterogeneity in human populations lowers the overall thresholds as a result of weak migration between patches.

# 1. Introduction

The existence of a threshold behaviour in the spread of infectious disease was first established by Kermack & McKendrick [1,2]. In

most cases the threshold can be expressed in terms of a single parameter, namely, the basic reproduction number, which has to be greater than unity for the disease to invade and/or persist in the population. This threshold translates to having a threshold population of susceptible hosts. On the other hand, a class of plant and animal infectious diseases (including human neglected tropical diseases, NTDs) [1,3–6] exhibit multiple thresholds, namely, a threshold susceptible host density together with a threshold initial infection level in the population [4–7]. Multiple thresholds are created via mechanisms ranging from social differences in susceptibles [8], partially efficient vaccination programmes [9,10], the structure of interaction multi-groups in HIV [11], density dependence in parasite dynamics [12] and asymmetry in death of susceptibles and infectious [7,8]. For the vector-borne disease visceral leishmaniasis (VL), which is primarily transmitted by bites of sandflies [13,14], the frequency-dependent contact structure together with disease-induced death rate generates two thresholds and it translates into a threshold sandfly population and an initial prevalence among human hosts. It is further known that the threshold criterion for transmission and extinction of a disease may show inter/intra-location variations [12].

The success of global efforts to control and eliminate infectious diseases such as VL, dengue, malaria, lymphatic filariasis and other NTDs necessitates understanding their invasion, persistence and extinction dynamics and thresholds. This becomes even more important as the implementation of the control programmes reduce the prevalence of these diseases to the WHO-set respective end targets [15–18]. In particular, the VL control programmes implemented in the Indian subcontinent are well into their maturity and are becoming more aggressive, yet there is the continued transmission. This raises the question as to how relevant is the WHO end target threshold of 1-in-10 000 cases at the sub-district level [19,20]? This question is more pertinent given the possibility of transmission-related heterogeneities at finer scales viz.: spatial heterogeneity and movement of humans, heterogeneity in biting of sandflies, availability of additional reservoirs for sandflies, implementation of control strategies such as vector control and treatment and finally post-kala-azar-dermal-leishmaniasis (PKDL) cases acting as reservoirs of parasites in treated cases.

The presence of additional hosts for sandflies near human dwellings [21,22] and the human PKDL cases act as persistent reservoir for VL parasites [21,23]. Host diversity affects the transmission dynamics in many vector-borne infectious diseases including leishmaniasis [24], Chagas disease [25], West Nile virus [26,27], malaria [28], yellow fever [29], Lyme diseases [30] among others. In the case of VL, control measures such as indoor residual spraying (IRS) can lead to changes in vector biting behaviours and a spillover to additional reservoirs. The concern of vector bites spilling over to other hosts is strengthened due to reports of additional reservoirs in India [31]. A relatively extensive understanding of transmission dynamics would inadvertently require deciphering the role of non-human reservoirs, host competences, feeding preference of vectors for human and non-human reservoirs and their relative abundances, along with human movement. On the other hand, although treatment of VL cases eliminates the fatal stage of the disease, some treated cases lead to PKDL, a known reservoir for parasites [13]. These factors add to the ecological complexity of the disease, ultimately affecting disease elimination thresholds and its resilience—the ability of the disease to persist under perturbation in epidemiological and ecological conditions [12,32,33]. As an example, sensitive epidemiological and ecological parameters affecting the thresholds include the biting, transmission, treatment, PKDL and spatial heterogeneity among others.

Mathematical models explored in the literature have considered human–sandfly interaction as well as other animal reservoirs, such as dogs (in Brazil for example [14,20,34,35]), as part of the VL transmission cycle. Most of these studies assumed that vectors were getting parasites from only one host [14,20,34,35]; however, in reality the VL disease system is made complex due to bites of multiple reservoirs [31] (goats and cows) driven by host competence and vector feeding preferences. On the other hand, following the abundance of 80/20 rule (Pareto principle) in biological systems [36], a heterogeneous sandfly biting behaviour is conceivable although as yet unexplored. Furthermore, the behaviour of thresholds in a multi-host setting driven by host competence and vector feeding preferences has also not been investigated in detail [27] vis-à-vis the WHO end-goal infection threshold of 1-in-10 000 at district or sub-district level.

In this paper, we consider a multi-host model of VL, where each host is relevant in the transmission cycle, and incorporate vector feeding preferences, biting heterogeneity and host competences. We then use the parameter estimates for the Indian subcontinent from the literature [14,20] as an example and demonstrate that two thresholds are created via the backward bifurcation (BB) and study the impact that vector feeding preference, host competence, relative host abundance, biting heterogeneity and control efforts (such as treatment) have on the two thresholds. We also investigate the resilience of the

disease-free and endemic states, i.e. their ability to persist under perturbation in epidemiological and ecological conditions, to the changing threshold conditions. In the next section, we describe the hierarchical multi-host model, followed by calculation of system thresholds, their resilience and sensitivity to parameter variations.

# 2. Material and methods

## 2.1. The model

### 2.1.1. Model outline

We assume that the VL transmission cycle consists of humans, two non-human reservoirs and sandflies. This assumption encapsulates the generality of a multi-reservoir–human–sandfly transmission because it captures the effects of competing reservoirs in terms of biting preferences, host competences, lifespan and relative population levels, which is otherwise not possible when only one reservoir is present in addition to humans. Further justification for considering two reservoirs follows from the fact that goats and cows are the most probable reservoirs living near human dwellings especially in India [31]. Additionally, the movement of humans and sandflies between cities/villages/towns (called patches) can also affect the transmission dynamics [37].

We model the transmission dynamics by assuming that humans, sandflies and other non-human reservoirs live on $n$-patches and only humans can move between different patches. Sandfly movement is ignored as their flight range is limited (approx. 300 m) [37]. Although human movement happens on two levels (i) short-term commutes (affecting transmissibility based on visitation to other patches) where their patch identity is preserved [38] and (ii) long-term movement which includes migration to other patches (affecting population dynamics) [38–41], here we assume only short-term movement between patches. The susceptible humans ($S_{Hi}$) of patch $i$ get bitten by infectious sandfly vectors ($I_{vj}$) while visiting a patch $j$, remain asymptomatic ($E_{Hi}$) for a period of time before becoming infectious $I_{Hi}$. Untreated infectious humans die because of infection or recover due to immunity, or get hospitalized ($T_{Hi}$) and receive treatment. Treated humans either recover or get PKDL ($K_{Hi}$) due to treatment failure. Humans with PKDL recover at a certain rate. Recovered humans are classified into ($R_{Hi}$). Recovered humans can lose immunity and join the susceptible class again. The two non-human reservoir populations are susceptible ($S_{Ai}$, $S_{Bi}$) and become infectious ($I_{Ai}$, $I_{Bi}$) by bites of infectious sandflies (ignoring the asymptomatic compartments due to lack of sufficient data) in patch $i$, lose immunity and join their respective susceptible classes again (SIS). The susceptible sandfly vectors in patch $i$ ($S_{vi}$) bite an infectious/asymptomatic/PKDL human (either resident or visiting patch $i$) or an infectious member of non-human reservoir, become latent ($E_{vi}$) and finally infectious ($I_{vi}$). The resulting sandfly–human–non-human interaction network and its corresponding flow chart is shown in figure 1. The equations describing the full $n$-patch spatial model are as follows:

$$\text{Humans} \begin{cases} \dfrac{\mathrm{d}S_{Hj}}{\mathrm{d}t} = \Lambda_{Hj} - \lambda_{Hj}S_{Hj} + \delta_{Hj}R_{Hj} - \mu_{Hj}S_{Hj}, \\[2mm] \dfrac{\mathrm{d}E_{Hj}}{\mathrm{d}t} = \lambda_{Hj}S_{Hj} - \mu_{Hj} + \sigma_{Hj}E_{Hj}, \\[2mm] \dfrac{\mathrm{d}I_{Hj}}{\mathrm{d}t} = f_j\sigma_{Hj}E_{Hj} - (\tau_{Hj} + \gamma_{Hj} + d_{Hj} + \mu_{Hj})I_{Hj}, \\[2mm] \dfrac{\mathrm{d}T_{Hj}}{\mathrm{d}t} = \gamma_{Hj}I_{Hj} - (r_{Tj} + \mu_{Hj})T_{Hj}, \\[2mm] \dfrac{\mathrm{d}K_{Hj}}{\mathrm{d}t} = \eta_j r_{Hj}T_{Hj} - (r_{Kj} + \mu_{Hj})K_{Hj} \\[2mm] \dfrac{\mathrm{d}R_{Hj}}{\mathrm{d}t} = (1 - f_j)\sigma_j E_{Hj} + \tau_{Hj}I_{Hj} + (1 - \eta_j)r_{Tj}T_{Hj} + r_{Kl}K_{Hj} - (\mu_{Hj} + \delta_{Hj})R_{Hj}, \end{cases} \qquad (2.1)$$

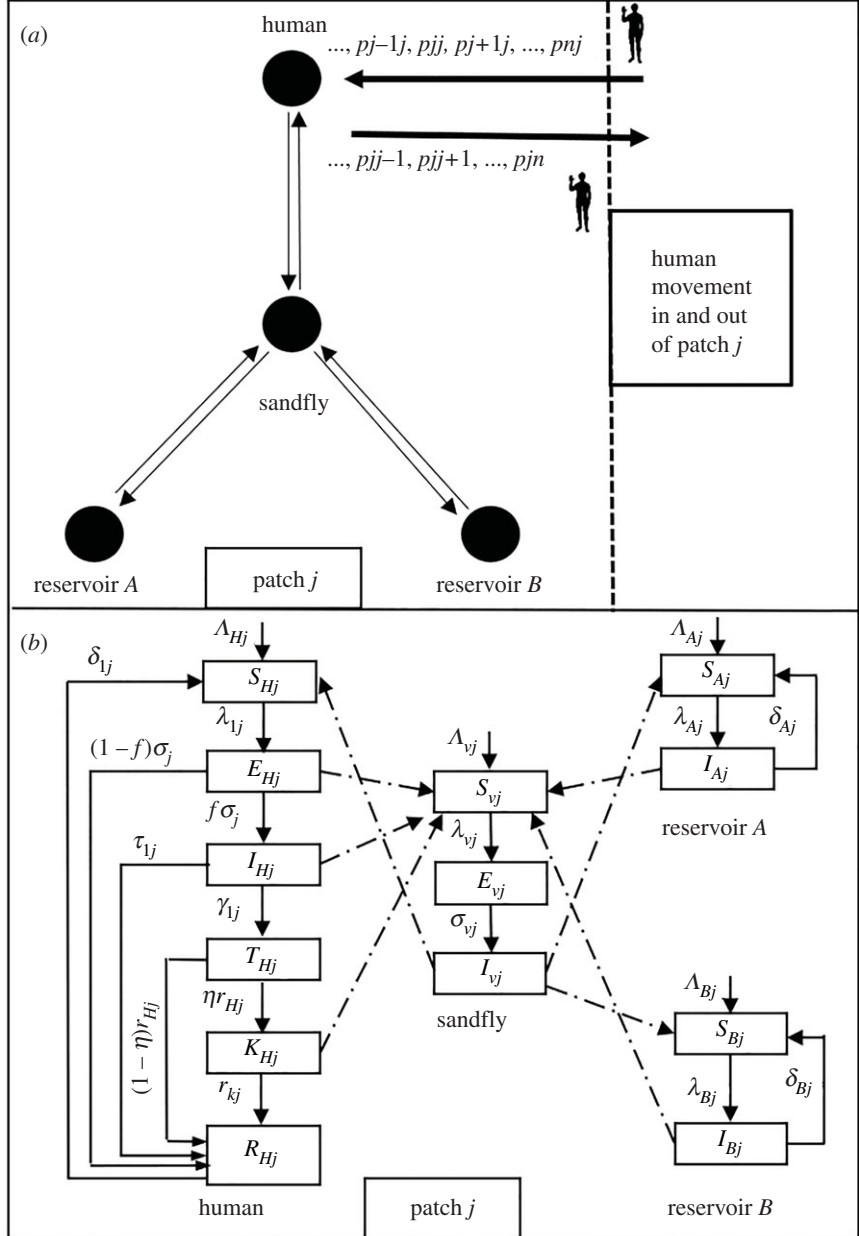

**Figure 1.** (*a*) The network of interaction formed by the transmission dynamics in visceral leishmania in patch *j*. (*b*) Flow diagram of the model in patch *j* as implied by equations (2.1)–(2.4).

$$\text{Reservoir } A \begin{cases} \dfrac{\mathrm{d}S_{Aj}}{\mathrm{d}t} = \Lambda_{Aj} - \lambda_{Aj}S_{Aj} + \delta_{Aj}I_{Aj} - \mu_{Aj}S_{Aj} \\[2mm] \dfrac{\mathrm{d}I_{Aj}}{\mathrm{d}t} = \lambda_{Aj}S_{Aj} - (\delta_{Aj} + \mu_{Aj})I_{Aj}, \end{cases} \tag{2.2}$$

$$\text{Reservoir } B \begin{cases} \dfrac{\mathrm{d}S_{Bj}}{\mathrm{d}t} = \Lambda_{Bj} - \lambda_{Bj}S_{Bj} + \delta_{Bj}I_{Bj} - \mu_{Bj}S_{Bj} \\[2mm] \dfrac{\mathrm{d}I_{Bj}}{\mathrm{d}t} = \lambda_{Bj}S_{Bj} - (\delta_{Bj} + \mu_{Bj})I_{Bj}, \end{cases} \tag{2.3}$$

$$\text{Sandflies} \begin{cases} \dfrac{\mathrm{d}S_{vj}}{\mathrm{d}t} = \Lambda_{vj} - \lambda_{vj}S_{vj} - \mu_{vj}S_{vj} \\[2mm] \dfrac{\mathrm{d}E_{vj}}{\mathrm{d}t} = \lambda_{vj}S_{vj} - (\sigma_{vj} + \mu_{vj})E_{vj} \\[2mm] \dfrac{\mathrm{d}I_{vj}}{\mathrm{d}t} = \sigma_{vj}E_{vj} - \mu_{vj}I_{vj} \end{cases} \qquad (2.4)$$

and

$$\left. \begin{aligned} \lambda_{lj} &= \sum_{i=1}^{n} k_i \log\left(1 + \frac{b\beta_l\alpha_l p_{ij}}{k_i\sum_l \alpha_l N_{li}} I_{vi}\right), \quad l = H, \\[2mm] \lambda_{lj} &= \sum_{i=1}^{n} k_i \log\left(1 + \frac{b\beta_l\alpha_l \Delta_{ij}}{k_i\sum_l \alpha_l N_{li}} I_{vi}\right), \quad l = A,B, \\[2mm] \lambda_{vj} &= \sum_{i=1}^{n} k_i \log\left(1 + \frac{b\beta_v[\alpha_H C_H p_{ji}(I_{Hi} + \rho_{1i}E_{Hi} + \rho_{2i}K_{Hi}) + \Delta_{ji}\alpha_A C_A I_{Ai} + \alpha_B \Delta_{ji}C_B I_{Bi}]}{k_i\sum_l \alpha_l N_{li}}\right) \\[2mm] \Delta_{ij} &= \begin{cases} 1 & i = j \\ 0 & i \neq j \end{cases}, \end{aligned} \right\} \qquad (2.5)$$

and   $i,j = 1, \ldots, n.$

In equation (2.5) index $l$ indicates human (denoted by $H$), and non-human reservoirs (denoted by $A$ and $B$), indices $i$ and $j$ are used for space discretization indicating different patches, $\Delta_{ij}$ is a notation used for the elements of matrix $\mathbb{P}$ used to denote residence times of non-human host in different patches (for example, $\Delta_{ij} = 0$ indicates that non-human hosts do not move between different patches, $p_{ij}$ are the elements of residence times matrix $\mathbb{P}$ capturing the strength of connectivity between patches $i$ and $j$ due to human movement). The symbols $\Lambda_{\blacksquare j}$, $\mu_{\blacksquare j}$ are used to represent recruitment and the natural death rate, respectively, of human, non-human hosts and sandfly vectors in the patch $j$. The symbol $\beta$ is used to represent transmission probability in humans, non-humans and vectors. The symbols $k_\blacksquare$ represent patch specific within host biting heterogeneity (more on this in subsections 2.1.2 and 2.1.3). Additional human parameters in patch $j$ are: $1/\sigma_{Hj}$ is the latency period, $\tau_{Hj}$ is the natural recovery rate, $d_{Hj}$ is the disease-induced death rate, $\gamma_{Hj}$ is the rate of seeking treatment/hospitalization in the infectious stage, $r_{Tj}$ is the rate of recovery/discharge following treatment/hospitalization, $\eta_j$ is the fraction of treated exhibiting PKDL, $r_{Kj}$ is the rate of recovery from PKDL stage. The parameter $\delta_{lj}$ is the loss of immunity for human and non-human hosts in patch $j$, $1/\sigma_{vj}$ is the latency period of sandfly vectors in patch $j$, $N_{lj}$ is the population of human and non-human hosts in patch $j$. The symbols $C_\blacksquare$ represent host competences, and $\alpha_\blacksquare$ represent vector biting preferences for each host.

In an isolated patch, biting heterogeneity in sandfly vectors on hosts may arise due to a limited flight range within the patch. Uniform homogeneous biting is recovered in equation (2.5) (also see electronic supplementary material, Document) for large values of $k_j$, the biting heterogeneity parameter in patch $j$. However, as $k_j$ gets closer to zero, the degree of heterogeneity in biting increases [42,43]. On the other hand, sandfly vectors may have different feeding preferences for humans and reservoirs $A$ and $B$. The 'vector feeding index' $\alpha_{lj}$ assesses the proportion of blood meals from host-$l$ in relation to the proportional abundance of host-$l$ in the host community [27], in general, it could be a complicated function of host population proportions [44]; however, here we take them as constant parameters. Additionally how competent are humans and reservoirs in transmitting the infection to susceptible sandflies is governed by *host competence* parameter, $C_{lj}$, defined as the ability of host-$l$ to successfully transmit the virus/parasite to the biting vector [45]. For simplicity, we assume host competency and vector biting preference do not change from patch to patch, i.e. $\alpha_{lj} = \alpha_l$ and $C_{lj} = C_l$ for all patches $j$. Note, in our model we allow for hosts to be dead-end ($C_l = 0$) as well as competitive hosts ($0 < C_l \leq 1$). A detailed list of parameters and their interpretation is given in table 1. A brief summary and description of the different types of heterogeneity and the force of infection terms in equation (2.5) are provided in subsections 2.1.2 and 2.1.3, respectively, and also summarized in table 2.

**Table 1.** Parameters and their interpretation in the human, two reservoirs and vector model equations (2.1)–(2.4).

| parameter | interpretation | units | numerical values |
|---|---|---|---|
| $\Lambda_S$ | sandfly birth rate | day$^{-1}$ | variable |
| $\Lambda_H$ | human birth rate | day$^{-1}$ | 4 801 062/(68.1 $\times$ 365) (Muzaffarpur human population 4 801 062) |
| $\Lambda_A$ | reservoir $A$ birth rate | day$^{-1}$ | 12 000/(10 $\times$ 365) (variable) |
| $\Lambda_B$ | reservoir $B$ birth rate | day$^{-1}$ | 315 656/(12 $\times$ 365) (variable) |
| $\mu_S$ | sandfly death rate | day$^{-1}$ | 1/14.0 (variable) |
| $\mu_H$ | human death rate | day$^{-1}$ | 1/(68.1 $\times$ 365) (68 years life expectancy in Muzaffarpur |
| $\mu_A$ | reservoir $A$ death rate | day$^{-1}$ | 1/(10 $\times$ 365) (life of goats: 10 years) |
| $\mu_B$ | reservoir $B$ death rate | day$^{-1}$ | 1/(12 $\times$ 365) (life of dogs/ cows: 12 years) |
| $1/\sigma_v$ | sandfly incubation period | day | 5 days |
| $1/\sigma_H$ | human incubation period | day | 166 days |
| $\tau_H$ | human recovery rate if left untreated | day$^{-1}$ | 0.00476 |
| $\delta_H$ | human rate of loss of immunity | day$^{-1}$ | 0.000913 (3 years) |
| $\delta_A$ | reservoir $A$ rate of loss of immunity | day$^{-1}$ | assumed to be equal to humans |
| $\delta_B$ | reservoir $B$ rate of loss of immunity | day$^{-1}$ | assumed to be equal to humans |
| $f$ | fraction of asymptomatic humans moving to infectious state | unitless | 0.14 (one in seven asymptomatics become infectious) |
| $\gamma_H$ | rate of reporting for treatment in humans | day$^{-1}$ | 0.001 (variable) |
| $r_T$ | human recovery rate after reporting for treatment | day$^{-1}$ | 1/30.0 (one month recovery period) |
| $\eta$ | fraction of humans moving to PKDL state after treatment | unitless | 0.2 (one in 20 VL treated people proceed to have PKDL) |
| $r_K$ | human recovery rate from PKDL | day$^{-1}$ | 1/547.5 (18 months treatment for PKDL) |
| $\alpha_H$ | vector biting preference for humans | unitless | variable |
| $\alpha_A$ | vector biting preference for reservoir $A$ | unitless | variable |
| $\alpha_B$ | vector biting preference for reservoir $B$ | unitless | variable |
| $C_H$ | human competence | unitless | [0,1] |
| $C_A$ | reservoir $A$ competence | unitless | [0,1] |
| $C_B$ | reservoir $B$ competence | unitless | [0,1] |
| $\beta_S$ | transmission probability from all reservoirs including humans to vector | unitless | 0.158 |
| $\beta_H$ | transmission probability from sandfly to human | unitless | 0.74 |
| $\beta_A$ | transmission probability from sandfly to reservoir $A$ | unitless | 0.74 |
| $\beta_B$ | transmission probability from sandfly to reservoir $B$ | unitless | 0.74 |
| $b$ | biting rate of sandflies on hosts | day$^{-1}$ | 0.25 |
| $\rho_1$ | fraction of asymptomatic humans contributing to infection in sandflies | unitless | 0.01 (variable) |
| $\rho_2$ | fraction of humans with PKDL contributing to infection in sandflies | unitless | 0.01 (variable) |

(*Continued.*)

| parameter | interpretation | units | numerical values |
|---|---|---|---|
| $d_H$ | disease-induced death rate in humans | day$^{-1}$ | 0.009 (variable) |
| $d_A$ | disease-induced death rate in reservoir $A$ | day$^{-1}$ | 0 (assumed) |
| $d_B$ | disease-induced death rate in reservoir $B$ | day$^{-1}$ | 0 (assumed) |
| $k_j$ | sandfly biting heterogeneity in patch $j$ | unitless | variable |
| $p_{ij}$ | elements of residence times matrix for humans between patch $i$ and $j$ | unitless | variable |
| $\Delta_{ij}$ | elements of residence times matrix for non-human reservoirs between patch $i$ and $j$ | unitless | 0 $i \neq j$  1 $i = j$ |
| $N_{lj}$ | population of human ($I = H$), reservoir $A$ ($I = A$) and reservoir $B$ ($I = B$) | unitless | |

**Table 2.** Heterogeneities in the model.

| heterogeneity | implementation |
|---|---|
| host and vector | separate dynamical equations for hosts and vectors |
| vector biting heterogeneity | negative binomial (see electronic supplementary material, Document) |
| human and reservoir | hosts are further classified into humans and reservoirs |
| reservoir heterogeneity | reservoirs $A$ and $B$ account of multiplicity of reservoirs |
| vector biting preference | biting preference for hosts and reservoirs; host and reservoir competence in transmitting infection to sandfly vectors |
| spatial heterogeneity | spatial heterogeneity is introduced by residence times matrix $\mathbb{P}$ |

### 2.1.2. Heterogeneities in the model

As mentioned in the previous section, the model (equation (2.1) to equation (2.5)) incorporates different types of heterogeneities and they are listed below:

1. different types of hosts (e.g. human, vector and animal reservoirs),
2. distinct reservoirs based on host competence,
3. spatial population distribution and structure, and
4. sandfly differential biting preferences.

Apart from human and sandfly vectors, we have considered two types of host reservoirs potentially representing animals. The efficiency of reservoirs to successfully transfer pathogens during a bite may vary depending on host competence. We assume that vector may have differential biting preference towards different host species and individuals in the population. The model captures discrete space via distinct population structure and movement patterns.

Many mathematical models in the literature primarily assume *homogeneous* biting patterns between hosts. In our model, we consider differential biting preferences towards different hosts, hence, capturing heterogeneity in biting rates.

### 2.1.3. The force of infection ($\lambda_\blacksquare$) in the model

The force of infection in equation (2.5), is defined as the *per capita* rate at which susceptible hosts/vectors contract the infection [46]. The force of infection on humans, reservoirs and sandfly vectors is derived by combining the short-term movement (Lagrangian movement [38]), within host sandfly biting

heterogeneity ($k_j$) [42,43], vector feeding indices $\{\alpha_{lj}\}$ [27] and host competences $\{C_{lj}\}$. Feeding preference assess the proportion of blood meals sandfly vectors have from host-$l$, whereas the biting heterogeneity $k_j$ indicates that all sandflies do not bite individuals in the host population equally and also that within similar host types all members may not receive the same number of sandfly bites.

First, consider a modelling scenario with an isolated patch. Sandfly biting heterogeneity in a patch has two components: first, biting preferences for reservoirs and humans vary (because of sandfly feeding behaviour). Second, the members of the same host (humans or each of the two reservoirs) get varying number of bites (because of host individual's characteristics). Most models in the literature assume that the sandfly–host encounters are well mixed, i.e. any given host (human or non-human reservoir) receives the same number of sandfly bites. In general, this assumption is unrealistic, as a given sandfly may only have the opportunity to bite a limited number of hosts (e.g. some humans may receive more bites than others) in the population. Following the formalism developed for dengue [42,43,47], we assume that the number of VL transmission causing bites on $m$th host are Poisson distributed with mean $\theta_m$. If all host members receive the same bites on average, then $\theta_m$ is the same for all the hosts in the population, implying the standard homogeneous transmission model. However, the biting means themselves may have a probability distribution over the population and under the assumption that the means $\theta_m$ follow a Gamma distribution (as mean bites are greater than zero and can take a large range of values) with shape parameter $k_j$ and a rate parameter $s_c$ (scale parameter $1/s_c$). Consequently, the marginal probability distribution of effective risky bites follows a negative binomial with mean $k_j/s_c = (b\beta_l\alpha_l/\sum_l \alpha_l N_{lj})I_{vi}$ (using composition of functions formula, Gamma distribution of Poisson distribution). With a few steps of calculation using the negative binomial [42,43,47], the risk of susceptibles becoming infectious is $1 - \left(1 + (b\beta_l\alpha_l/k_j \sum_l \alpha_l N_{lj})I_{vi}\right)^{k_j}$ (details are given in the electronic supplementary material, Document). Using the relationship between risk and rate [48] we obtain the rate of infection of susceptible hosts as $\lambda_{lj} = k_i \log\left(1 + (b\beta_l\alpha_l/k_j \sum_l \alpha_l N_{lj})I_{vi}\right)$. Through a similar argument, the rate of infection for susceptible sandflies after biting infectious humans/reservoirs is $\lambda_{vj} = k_j \log\left(1 + (b\beta_v[\alpha_H C_H(I_{Hj} + \rho_{1j}E_{Hj} + \rho_{2j}K_{Hj}) + \alpha_A C_A I_{Aj} + \alpha_B C_B I_{Bj}]/k_j \sum_l \alpha_l N_{lj}0)\right)$. In these expressions $k_j$ characterizes the level of heterogeneity of bites (susceptible humans/reservoirs by infectious sandfly vectors and infectious humans/reservoirs by susceptible vectors) [42,43]. Finally, incorporating the movement of humans between $n$-patches, equation (2.5) is obtained.

In this study, we study a *one-patch model* in relative detail and present preliminary results for a two-patch model separately. In the one-patch model, we increase its complexity as follows: first, switching off the parameters related to treatment and PKDL and other reservoirs we obtain a *base model* (BM; considered as our first model); second, including human treatment, we get the *treatment model* (BMT); third, with both *treatment and PKDL* parameters, the BMT-PKDL model is obtained. Finally, we *include reservoirs* in the transmission cycle. We will work our way upwards systematically adding complexity as described and later we will study a *two-patch model* with only humans and sandflies in the transmission cycle. In the subsequent sections, we will assume homogeneous biting and obtain system thresholds for simplicity of algebraic calculations. We come back to discussing the implications of heterogeneous biting on system thresholds in the Results and Discussion and conclusion sections.

## 2.2. Analysis

In this section, we assume homogeneous biting ($k_j$ is large so that the logarithmic term in equation (2.5) reduces to the standard mass action force of infection [46]) and obtain threshold conditions for the model.

### 2.2.1. Thresholds and multiple equilibria in one-patch model

In our model $R_0$ acts as one of the thresholds and it can be translated to a threshold biting or threshold sandfly population. This means that VL will invade the population only if $R_0 > 1$ [49]; however, if the disease-induced death rate is sufficiently strong then it can invade and persist even at a lower value of $R_0$, i.e. $R_0 \geq R_c \leq 1$, provided that the initial proportion of the infectious population is above the infection threshold. The infection threshold serves as an additional threshold and occurs due to a phenomenon known as BB. That this phenomenon is possible in VL is highlighted from the broad spectrum of parameter values estimated in previous studies on VL in the Indian subcontinent. In particular, one such study reported extra deaths caused by VL [14,20] which we find responsible to generate multiple equilibria and therefore multiple thresholds via BB.

The system of equations (2.1)–(2.4) exhibit two stable and one unstable infectious state under a restrictive parameter regime if the disease-induced death is sufficiently strong. The unstable infectious

states act as a threshold infection prevalence, above (below) which the VL invades (goes extinct in) the population. Another threshold arises out of a minimum sandfly population: below this sandfly number VL cannot persist in the population regardless of the initial prevalence of VL. Next we obtain expressions for these thresholds and discuss the regimes of their validity.

Assuming homogeneous biting, the model equations (2.1)–(2.4) have a disease-free equilibrium (DFE) given by

$$X_{0j} = \left( \frac{\Lambda_{Hj}}{\mu_{1j}}, 0, 0, 0, 0, 0, \frac{\Lambda_{Aj}}{\mu_{Aj}}, 0, \frac{\Lambda_{Bj}}{\mu_{Bj}}, 0, \frac{\Lambda_{vj}}{\mu_{vj}}, 0, 0 \right). \tag{2.6}$$

The basic reproduction ratio $R_0$ evaluated at the DFE using the next generation matrix approach is

$$\left. \begin{aligned} R_0^2 &= \left( \frac{b^2 \beta_v \sigma_v N_v^0}{\mu_v Q_v N_{tot}^0} \right) \sum_{l=\{1,A,B\}} \frac{\beta_l \alpha_l m_l N_l^0}{N_{tot}^0}, \\ m_1 &= \frac{\alpha_H C_H (Q_3 Q_4 (f\sigma_H + \rho_1 Q_2) + \rho_2 \eta r_T \gamma_H f \sigma_H)}{Q_1 Q_2 Q_3 Q_4}, \\ m_A &= \frac{\alpha_A C_A}{Q_6}, \\ m_3 &= \frac{\alpha_B C_B}{Q_7} \\ N_{tot}^0 &= \sum_{l=\{H,A,B\}} \alpha_l N_l^0, \end{aligned} \right\} \tag{2.7}$$

and

where $Q_{1j} = \mu_{Hj} + \sigma_{Hj}$, $Q_{2j} = \tau_{Hj} + \gamma_{Hj} + d_{Hj} + \mu_{Hj}$, $Q_{3j} = r_{Tj} + \mu_{Hj}$, $Q_{4j} = r_{Kj} + \mu_{Hj}$, $Q_{5j} = \mu_{Hj} + \delta_{Hj}$, $Q_{6j} = \delta_{Aj} + \mu_{Aj}$, $Q_{7j} = \delta_{Bj} + \mu_{Bj}$, $Q_{vj} = \sigma_{vj} + \mu_{vj}$. The above expression for $R_0$ is true for the general one-patch model. The corresponding expression for the BM is recovered by setting $\gamma_H = \eta = r_T = r_K = \rho_2 = N_{A,B}^0 = 0$, $R_0$ for BMT is recovered by setting $\eta = r_K = \rho_2 = N_{A,B}^0 = 0$ and $R_0$ for BMT-PKDL is recovered by setting $N_{A,B}^0 = 0$. In this VL system, a threshold $R_0$ is translatable to a sandfly population threshold. In general, the VL will invade the population only if $R_0 > 1$ [49]; however, if the disease-induced death rate $d_H$ is sufficiently strong then it can invade and persist even at a lower threshold, i.e. $R_0 \geq R_c \leq 1$, provided that initial infection prevalence is above the infection threshold due to BB [50]. This invariably generates two thresholds in the system giving rise to a characteristic bi-stable region $R_c \leq R_0 \leq 1$ so that the DFE coexists with a stable and an unstable endemic equilibrium: the value of $R_c$ gives the threshold vector population ($N_{Sc}$) while the unstable endemic equilibrium forms the human infection prevalence threshold ($I_{Hc}$).

The endemic equilibria, and hence the human infection prevalence threshold ($I_{Hc}$), are easily obtained in terms of the force of infection on humans $\lambda_H$ which satisfies a sextic polynomial equation

$$A_6 \lambda_H^6 + A_5 \lambda_H^5 + A_4 \lambda_H^4 + A_3 \lambda_H^3 + A_2 \lambda_H^2 + A_1 \lambda_H^1 + A_0 = 0. \tag{2.8}$$

The steps to obtain the coefficients $\{A_i\}$ are given in the electronic supplementary material, Document. The endemic equilibria $X_1 = (S_H^*, E_H^*, I_H^*, T_H^*, K_H^*, R_H^*, S_A^*, I_A^*, S_B^*, I_B^*, S_v^*, E_v^*, I_v^*)$, force of infection on reservoir $A$ and reservoir $B$ are related to the force of infection on humans as

$$\lambda_A = \frac{\alpha_A}{\alpha_H} \lambda_H, \;\; \lambda_B = \frac{\alpha_B}{\alpha_H} \lambda_H, \tag{2.9}$$

$$\text{Humans} \left\{ \begin{aligned} S_H^* &= \frac{\Lambda_H}{(\lambda_H(1 - \delta_H F_1^H) + \mu_1)}, & E_H^* &= \frac{\lambda_H S_H^*}{Q_1}, \\ I_H^* &= \frac{f\sigma_H \lambda_H S_H^*}{Q_1 Q_2}, & T_H^* &= F_1^H \lambda_H S_H^*, \\ K_H^* &= \frac{f\sigma_H \lambda_H S_H^*}{Q_1 Q_2}, & R_H^* &= F_1^H \lambda_H S_H^*. \end{aligned} \right\} \tag{2.10}$$

$$\text{Res } A,B \left\{ \begin{aligned} S_A^* &= \frac{\Lambda_A}{(\lambda_A(1 - \delta_A F_1^A) + \mu_A)}, & I_A^* &= F_1^A \lambda_A S_A^* \\ S_B^* &= \frac{\Lambda_3}{(\lambda_B(1 - \delta_B F_1^B) + \mu_B)}, & I_B^* &= F_1^B \lambda_B S_B^* \end{aligned} \right\} \tag{2.11}$$

and

$$\text{Sandflies} \begin{cases} S_v^* = \dfrac{\Lambda_v}{(\lambda_v + \mu_v)}, & E_v^* = \dfrac{\lambda_v S_v^*}{Q_v} \\[2ex] I_v^* = \dfrac{\sigma_v \lambda_v S_v^*}{\mu_v Q_v}, & \end{cases} \tag{2.12}$$

where

$$F_1^H = \left[ \frac{(1-f)\sigma_H}{Q_1} + \frac{\tau_H \sigma_H f}{Q_1 Q_2} + \frac{(1-\eta) r_T \gamma_H f \sigma_H}{Q_1 Q_2 Q_3} + \frac{r_K \eta r_T \gamma_H f \sigma_H}{Q_1 Q_2 Q_3 Q_4} \right] \frac{1}{Q_5},$$

$$\boldsymbol{F_1^A} = \frac{1}{\boldsymbol{Q_6}}$$

and

$$\boldsymbol{F_1^B} = \frac{1}{\boldsymbol{Q_7}}.$$

In our model, the multiple thresholds are created (due to BB) if the disease-induced death rate parameter $d_H$ exceeds a certain value such that $\partial \lambda_1 / \partial R_0 |_{R_0=1} < 0$ in equation (2.8) [46], assuming $d_A = d_B = 0$. Note that the condition for BB is never satisfied if $d_H = 0$ as well. Given the multi-host nature of our model (and equation (2.8)) a closed-form relationship between the disease-induced death rate and other parameters for the existence of multiple thresholds to occur is cumbersome, therefore, we numerically confirmed that this criterion is satisfied for our choice of $d_H$.

### 2.2.2. System resilience and integral stability ($d_{IS}$)

We quantify the resilience of disease-free/endemic equilibria in the system using the integral stability index. Integral stability (IS) [33], $0 \leq d_{IS} \leq 1$, is a measure of the resilience of individual equilibria/ attractors in a multi-states-stable dynamical system. Resilience is defined as the capacity of a system to recover in the face of perturbations. Since in BB the endemic state coexists with a DFE, we characterize their resilience using the integral stability. For a detailed mathematical definition of IS see electronic supplementary material, Document.

### 2.2.3. Sensitivity of system thresholds

The system thresholds are sensitive to variations in parameters of the system. We use the formalism of partial rank correlation coefficient (PRCC) [51–53] to characterize the sensitivity of thresholds to variations in systems parameters. To calculate sensitivity to model parameters each parameter is assigned a Gaussian distribution with their respective mean as in table 1 and standard deviation to be 1/30 of the mean (so that neither of the thresholds are eliminated and therefore facilitating their sensitivity analysis). A more realistic assumption on the distribution of parameters will require the distribution of parameters to be calibrated against field data (which we do not attempt here).

### 2.2.4. Thresholds in two-patch model

To study the effect of spatial heterogeneity we consider the BM in two spatially isolated locations. Coupling between the patches is assumed to be only via human movement between them, making the simplifying assumption that sandflies do not move between patches. The threshold $R_{0S}$ for the two-patch system is given by

$$R_{0S} = \sqrt{\frac{1}{2}(R_{01}^2 + R_{02}^2 + R_{0h21}^2 + R_{0h12}^2 + R_{sqrt})}, \tag{2.13}$$

where $R_{0h21}$, $R_{0h12}$ and $R_{sqrt}$ are described in the electronic supplementary material, Document, $R_{01}^2$ and $R_{02}^2$ are thresholds for isolated patches. The expression for $R_{0S}$ shows that system can be above the global threshold even if threshold condition in isolated patch may not be satisfied. The threshold infection prevalence (discussed in the Results section) is obtained numerically using Newton–Raphson root finding method.

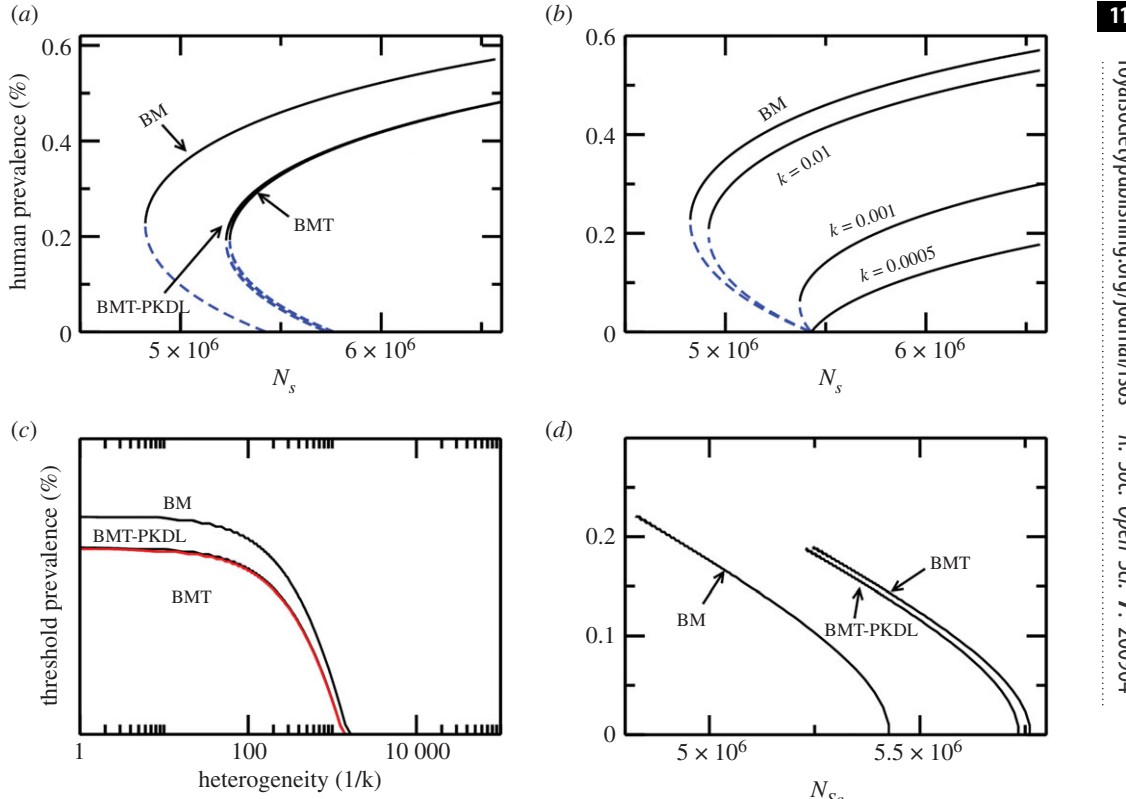

**Figure 2.** (a) Bifurcation diagrams showing the effects of treatment and subsequent PKDL on the system thresholds: when treatment is applied in the base model the system exhibits a new set of thresholds. Compared with the base model, when treatment is applied a higher sandfly threshold is obtained whereas the infection threshold is lowered. However, at the threshold sandfly population in the base model the disease is eliminated under treatment. PKDL occurs due to treatment failure, therefore, the thresholds lie in between the treatment and base models. (b) Bifurcation diagrams for different values of biting heterogeneity in the base model (BM) indicating that heterogeneity increases the sandfly threshold and lowers the infection threshold. (c) and (d) trace the threshold human infection prevalence and sandfly threshold as a function of biting heterogeneity for base model (BM) and with treatment and PKDL.

## 3. Results

In the one-patch BM with homogeneous biting (large $k$), i.e. no reservoir $A$ and reservoir $B$ (and no treatment in humans) the sextic polynomial, equation (2.8), reduces to the quadratic equation

$$A_2\lambda_H^2 + A_1\lambda_H^1 + A_0 = 0. \tag{3.1}$$

For a sufficiently large disease-induced death rate in humans [14,20] BB, with the characteristic two system thresholds, is observed: the first threshold is the human infection prevalence threshold (calculated using equations (2.10)–(2.12)) and the second threshold is the threshold vector population related to $R_c$ as obtained from the condition when $A_1^2 - 4A_2A_0 = 0$:

$$R_c = f(\Lambda_l, \mu_l, d_l, \Lambda_v, \mu_v, \ldots), \; l = \{H, A, B\} \tag{3.2}$$

here the function $f(\Lambda_l, \mu_l, d_l, \Lambda_v, \mu_v, \ldots)$ on the right-hand side of equation (3.2) depends on all the parameters of the system, see electronic supplementary material, Document for details. For the values of the parameters considered (table 1) human infection prevalence threshold for the BM is 0.2% at the threshold sandfly population (approx. $4.7 \times 10^6$) as shown in the bifurcation diagram for the BM as a function of vector population density $N_S$ in figure 2a.

When all the reservoir hosts (goats and cows) are also present then the sextic polynomial equation (2.8) is solved numerically to obtain the threshold vector population numbers $N_{Sc}$ ($R_c$). The presence of reservoirs comes with two possibilities:

1. non-human reservoirs are dead-end hosts, i.e. $C_A = C_B = 0$
2. at least one or all non-human reservoirs are competitive, i.e. $C_A$ or $C_B = 0$, $C_A = C_B = 1$.

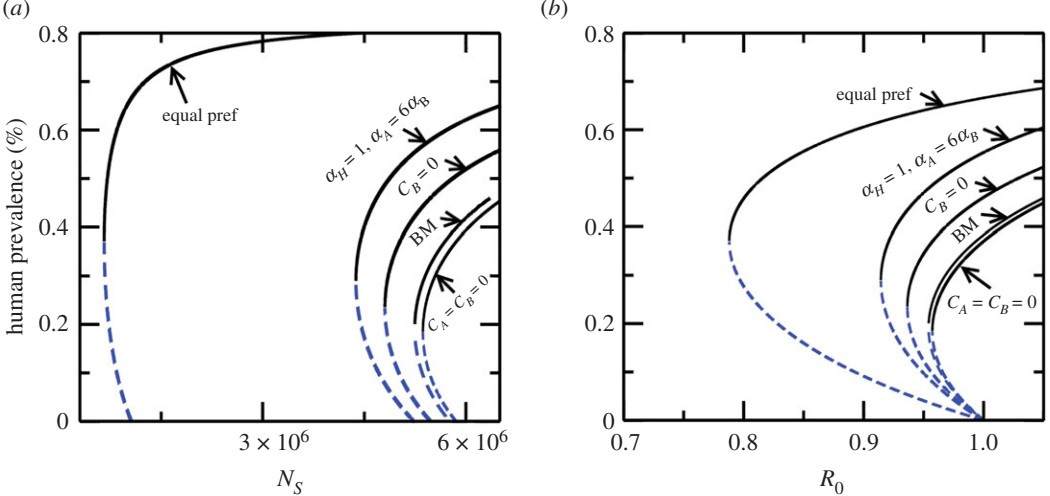

**Figure 3.** Bifurcation diagram showing how thresholds of the system change in presence of animal reservoirs (goats and cows) in addition to humans. (*a*) As a function of sandfly population (*b*) as a function of $R_0$. The bifurcation diagrams are obtained for various scenarios capturing host competence ($C_A$, $C_B$) and vector biting preference for goats and cows. The curves to the left of 'BM' correspond to cases of (i) only goats are competitive hosts ($C_H = 1$, $C_A = 1$, $C_B = 0$) with biting preferences of ($\alpha_H = 1$, $\alpha_A = 0.6$, $\alpha_B = 0.1$) (ii) goats and cows both are competitive hosts ($C_H = 1$, $C_A = 1$, $C_B = 1$) with biting preferences of ($\alpha_H = 1$, $\alpha_A = 0.6$, $\alpha_B = 0.1$) (iii) goats and cows both are competitive hosts ($C_H = 1$, $C_A = 1$, $C_B = 1$), however, with equal biting preferences ($\alpha_H = 1$, $\alpha_A = 1$, $\alpha_B = 1$). The curves to the right of 'BM' show the case when goats and cows are dead-end hosts with equal biting preferences ($\alpha_H = 1$, $\alpha_A = 1$, $\alpha_B = 1$). These curves show the dilution and amplification effects on the thresholds in presence of additional reservoirs with respect to the base model (when only humans are present).

To investigate the effects of biting preference, we consider the following cases: (i) when all hosts are equally competitive, i.e. $C_H = C_A = C_B = 1$, and sandflies prefer to bite them equally $\alpha_H = \alpha_A = \alpha_B = 1$, (ii) all hosts are equally competitive, i.e. $C_H = C_A = C_B = 1$ but $\alpha_H = 1$, $\alpha_A = 0.6$, $\alpha_B = 0.1$, (iii) cows are dead-end hosts, i.e. $C_H = C_A = 1$, $C_B = 0$ but sandflies prefer to bite humans and goats more as compared with cows $\alpha_H = 1$, $\alpha_A = 0.6$, $\alpha_B = 0.1$, and (iv) cows and goats are dead ends and biting preference is the same as in case (iii). In cases (i)–(iii) presence of reservoirs increases the bi-stable region, by first reducing the sandfly threshold ($N_{Sc}$) and increasing the human infection prevalence threshold, while in case (iv) the threshold ($N_{Sc}$) is larger and the human infection prevalence threshold is smaller relative to the BM, thereby reducing the bi-stable region, see figure 3*a*.

Another interesting feature that multiple hosts highlight is the changing resilience of the DFE/endemic state as the reservoirs are added to the system. The integral stability (IS) $d_{IS}$ shows that if bi-stable region increases (decreases) then the resilience of the endemic state increases (decreases). The increased (decreased) resilience can be explained by the observation that when bi-stable region is large (small) a smaller (larger) number of sandflies are needed to stabilize the endemic state. In addition, a smaller number of initial human infections can push the system to the endemic state for a given number of sandflies (above the sandfly threshold for the BM). In the case where other reservoirs are dead-end hosts, these two factors combine to show an increase in resilience ($d_{IS}$) of DFE, as shown in figure 4.

Controlling the spread of infection requires looking for mechanisms to stabilize the DFE, and when BB is present the traditional way of ensuring $R_0 < 1$ for stabilization of DFE does not work, as control in such circumstances requires keeping track of the two thresholds:

1. the minimum vector population threshold $N_{Sc}$
2. the threshold infection prevalence level in humans as defined by the unstable endemic equilibrium.

If either of these two thresholds are crossed from above, i.e. decreasing sandflies population to below $N_{Sc}$ and reduce human infection prevalence than the infection threshold, DFE is stabilized. When humans are subjected to treatment at a constant rate, both these thresholds shift such that the human infection threshold is reduced while increasing the threshold vector population $N_{Sc}$ for the disease to emerge. Humans with PKDL also transmit the disease resulting in an endemic state intermediate to BM and the model with a completely efficient treatment, as shown in figure 2*a*.

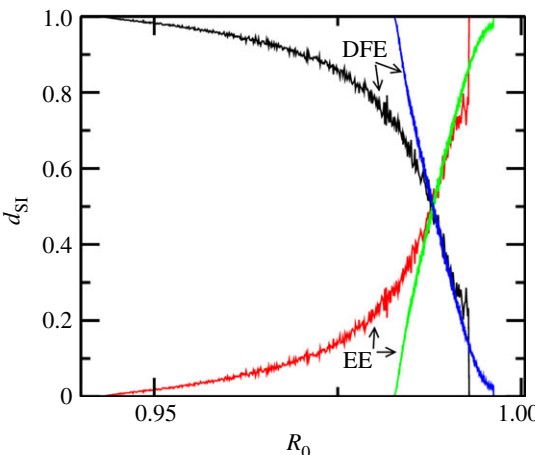

**Figure 4.** Integral stability of the DFE and endemic equilibria (indicated by arrows) for the base model (black and red) and multi-host models (blue and green). In the multi-host model, dead-end reservoirs were considered to generate this diagram, demonstrating that the DFE is more resilient when dead-end hosts are introduced. Based on this diagram and figure 3, it is easy to see that the endemic state is more resilient when competitive reservoirs are considered.

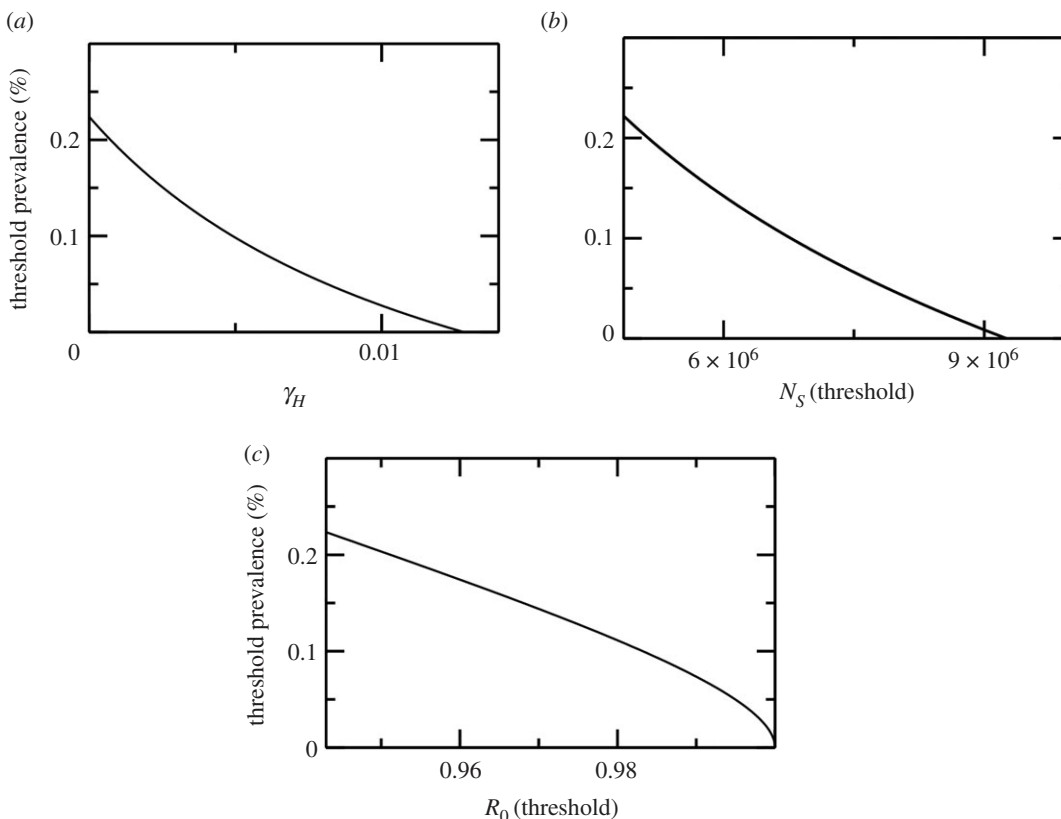

**Figure 5.** The human infection prevalence threshold as a function treatment rate $\gamma_H$ (in the absence of reservoirs). It shows that a cascade of prevalence threshold values are created as the treatment rate is increased, before eventually the prevalence threshold drops to zero as shown in (a). This leaves behind only the sandfly population threshold as shown in (b), equivalently only the threshold $R_0 = 1$ (c).

Although having less efficient treatment is better than no treatment at all. This implies that, assuming the vector population does not increase, treatment can eliminate the disease; however, if the vector population is allowed to grow then the disease can remain endemic. Thereby the goal of reaching the human infection threshold (or the threshold prevalence) via treatment alone would become ineffective as it cascades to zero with increasing treatment rates, as shown in figure 5. Similar shifts in thresholds are observed when treatment of human hosts or vector control is applied in presence of other

reservoirs; however, the multi-host system relatively amplifies (dilutes) the effect of control strategies when dead-end (competitive) reservoirs are present, making VL control either easier or more difficult relative to when humans are the only hosts.

To study the effects of biting heterogeneity on system thresholds we take the BM, and allow an increasingly heterogeneous sandfly biting behaviour by changing the parameter $k$. As heterogeneity increases in the system the sandfly threshold increases, implying that large number of flies are required for infection to emerge; however, the relatively large sandfly threshold also lowers the threshold human infection prevalence. The behaviour of thresholds follow similar trend when host complexity is increased by adding more disease stages as discussed above for the homogeneous mixing case; however, in comparison the prevalence level is reduced. The thresholds shift in two ways. First, the sandfly threshold is increased and the infection threshold is larger as compared with the BM. Second, at the new sandfly threshold the infection threshold is lower in comparison with the BM. A bifurcation diagram indicating these trends is shown in figure 2$b$.

The thresholds obtained for the BM are sensitive to changes in parameters. We evaluate the sensitivity to model parameters corresponding to the BM such as sandfly biting rate on humans, PKDL and other parameters using PRCC. Detailed tornado plots showing PRCC for threshold sandfly population ($N_{Sc}$) and human infection prevalence threshold ($I_{Hc}$) are shown in figure 6$a$,$b$. The PRCC was obtained by assuming Gaussian distribution for varied parameters, giving rise to a distribution of threshold ($N_{Sc}$, $I_{Hc}$) as shown in figure 6$c$–$e$. A positive/negative PRCC for a parameter means that the threshold increases/decreases with increasing/decreasing the parameter. The behaviour of PRCC shows that sandfly threshold ($N_{Sc}$) is strongly but negatively sensitive to biting rate ($b$), biting heterogeneity parameter ($k$), the factor of asymptomatic/latent moving to infectious class ($f$), transmission parameters from fly to human (vice versa) ($\beta_H$, $\beta_v$), natural mortality rate of humans ($\mu_H$) and proportion of infectious bites from PKDL cases ($\rho_2$). On the other hand it is strongly but positively sensitive to sandfly mortality ($\mu_v$), human birth rate ($\Lambda_H$) and disease-induced death rate ($d_H$). Similarly, the human infection prevalence threshold is positively sensitive to the factor of asymptomatic/latent moving to infectious class ($f$) and loss of immunity ($\delta_H$). The biting parameter ($b$) has a strong negative effect on this threshold as expected (if more bites, then infection can emerge at a lower infection prevalence). To control VL it is, therefore, desirable to enhance the parameters increasing the sandfly population threshold as well as increasing the human infection threshold for an optimal control of visceral leishmaniasis: as an example, PRCC indicates that the number of bites have to be decreased (so that a large number of sandflies are required for disease to emerge).

The movement of humans between neighbouring villages/towns can also affect the local thresholds. To study the effect of human movement between otherwise isolated locations we assume the BM in two patches. The coupling between the patches is due to short-term human movements, where humans stay for most time $p_{ii}$ in their home patch and spend a fraction $p_{ij}$ in the other patch, thus the fraction of time spent is the coupling strength (see equations (2.2)–(2.6) and their description). The spatial effect is revealed as soon as asymmetry in parameters of the models in two locations are introduced, as an example we chose the sandfly birth rate and human birth rate as asymmetric parameters and considered two different values of the coupling ($p_{ii}$, $p_{ij}$). Following this scheme, a bifurcation diagram demonstrating (2.3) effects of spatial coupling and parameter heterogeneity is shown in figure 7. In the first instance, we assume that birth rate of sandflies in patch one is same as that in a one-patch model discussed above, while in patch two it is 0.081 times that in patch one. We find that in presence of human movement, patch one exhibits higher sandfly threshold and higher infection prevalence threshold, whereas patch two has a lower sandfly threshold and lower infection prevalence threshold as compared with the isolated one-patch model. As coupling strength is increased the dynamics in the two patches is as follows: the sandfly threshold in patch one increases while infection threshold decreases, and in patch two the sandfly threshold decreases and so does the threshold infection prevalence (figure 7$b$). However, the effect of spatial coupling is to lower the sandfly threshold and infection prevalence threshold in the two patches when compared with an isolated patch. Similar effects are observed when asymmetry in human birth rate (and hence population) is also considered, as shown in figure 7$c$.

# 4. Discussion and conclusion

In this paper, we investigated the dynamics of visceral leishmaniasis via a dynamical system model that may exhibit multiple thresholds due to BB phenomenon in a certain parameter regime. BBs belong to a

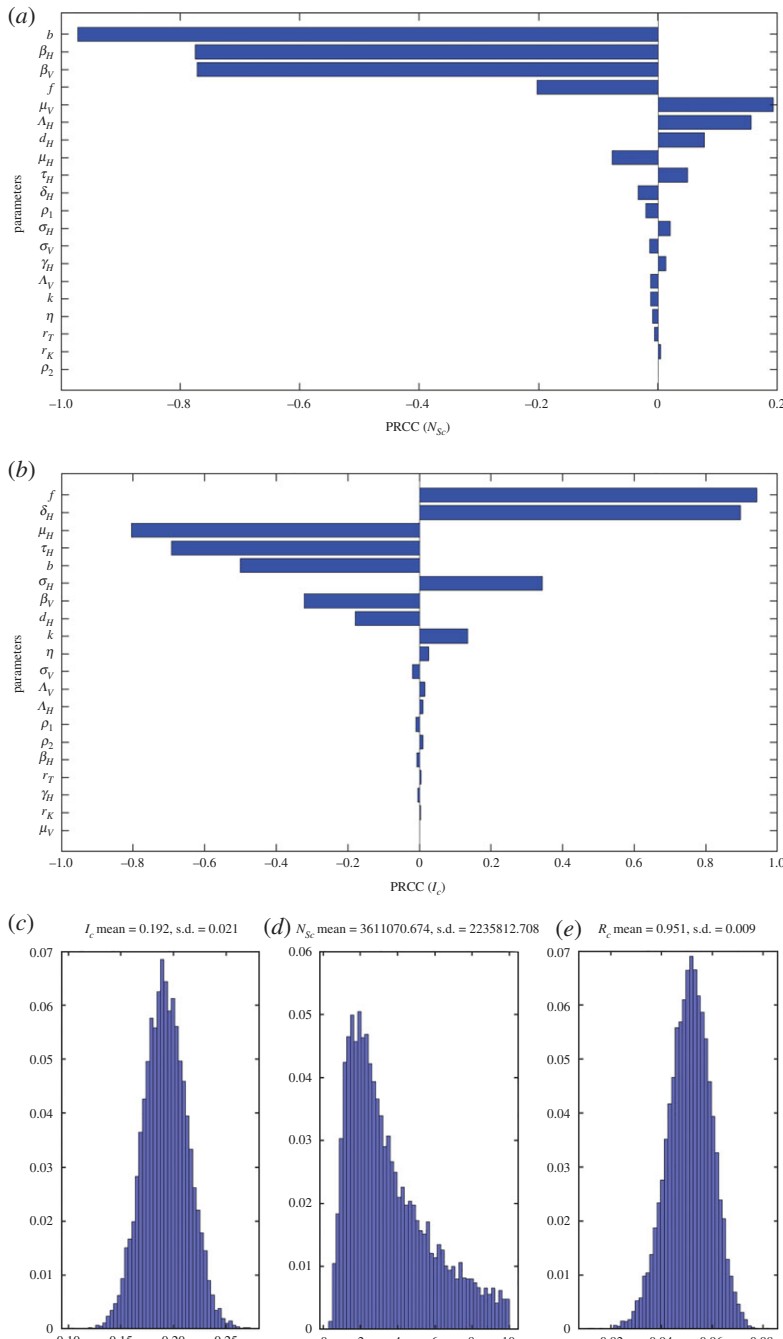

**Figure 6.** Tornado plot diagrams, (*a*) and (*b*), showing sensitivity of thresholds ($N_{Sc}$, $I_c$) on the *x*-axis, as calculated with the PRCC method, with respect to variations in system parameters on the *y*-axis. The distribution of thresholds ($N_{Sc}$, $I_c$, $R_c$) (*c*–*e*) as obtained from the distribution of parameters used for PRCC evaluation.

class of disease systems where, in addition to the basic reproduction number, $R_0$ (equivalently the threshold levels of vector bites/vector population), initial size of infected individuals in hosts act as a threshold. Such systems have been observed in both micro- and macro-parasitic systems such as, tuberculosis (TB), dengue, lymphatic filariasis and onchocerciasis [1,4–6,12,54,55]. Although the multi-threshold systems have been explored both in the field [55] as well as in theoretical literature [6], the theme which it has demonstrated consistently is that control is not trivial in systems exhibiting multiple thresholds. Our current study investigates dynamical regimes where VL might exhibit multiple thresholds; thus, a potential cause for presenting challenges in control of VL in the Indian subcontinent. This model exhibits multiple thresholds due to BB phenomenon driven by nonlinear relationship between human mortality (disease-induced and natural mortality rates) and sandfly

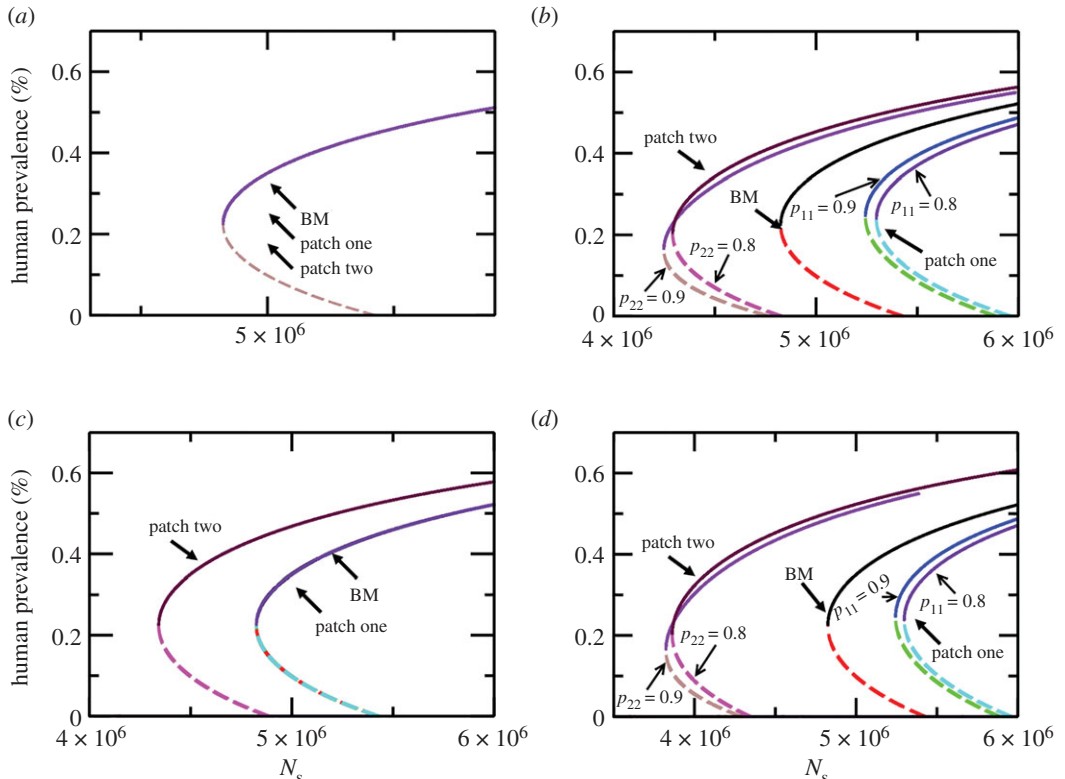

**Figure 7.** Bifurcation diagrams showing backward bifurcation in a two-patch model. Each patch assumes a base model (no treatment, no PKDL, no biting heterogeneity, no non-human reservoirs) with coupling between the two patches represented by the time humans spend in their home and neighbouring patch (see equations (2.1)–(2.4)). (*a*) Bifurcation diagram is the same as base model in isolated patches when each patch has identical demographic parameters for humans and sandflies, regardless of coupling strength. (*b*) Bifurcation diagram for different coupling strengths (0.8 and 0.9 as indicated by arrows) when sandfly birth rate (and subsequently sandfly population) is higher in patch one than patch two. (*c*) Bifurcation diagrams in the two patches when human birth rate is higher in patch one than patch two. (*d*) Bifurcation diagram for different coupling strengths (0.8 and 0.9 as indicated by arrows) in the two patches when human and sandfly birth rates are greater in patch one than patch two. These diagrams show that spatial heterogeneity in combination with parameter asymmetry can lower the thresholds as compared with an isolated system and the degree to which this happens depends on coupling strength.

transmission and demographic parameters (see electronic supplementary material, Document). Human infection prevalence spontaneously increases when the initial prevalence crosses a threshold level; however, there are two points to note: (i) new infections are generated by transmission from other population (sandfly population) even though disease-induced death in humans acts like a leakage in the system and (ii) increment in infection does not occur in human but in sandfly population, which then feeds back into human population. On the other hand, several reports have indicated the continued persistence of VL despite low death rates and no reported incidence in neighbouring regions [56,57], potentially as a result of higher number of initial infected arrivals in the region as compared with its neighbours. The initial infection prevalence has been recently understood to be a driving factor behind the emergence of chikungunya in South America [58,59], where despite international travel of people from affected regions for decades, the disease could not become endemic until a threshold of imported infections was reached in 2013. Importantly, the imported infection could not be accounted for in calculations of $R_0$, which would have been less than one because the disease was not present in the region in the first place. This draws similarities with BBs vis-à-vis the threshold prevalence and highlights the importance of investigating initial infection prevalence among other factors.

We began with a BM wherein only humans interact with sandflies without the influence of any control measures. Using the parameters from literature for India, where a sufficient amount of disease-induced death rate for humans was reported [14,20], we find that this BM exhibits BB. This presents a scenario where the system exhibits two thresholds: (i) human infection and (ii) sandfly

population density thresholds. Introducing treatment of infection in humans can help cross the human infection threshold, beyond which the disease is eventually eliminated; however, if sandfly population is left unchecked and continues to grow then human infection threshold presents itself as a cascading function of treatment, before eventually becoming zero, thereby making only a complete elimination of infection, as a rather impractical and only treatment goal. On the other hand, if the sandfly population remains the same, then the disease-free equilibrium can be stabilized when a sufficient rate of treating humans is employed. The scenario of increasing sandfly thresholds in conjunction with decreasing infection prevalence threshold can be equated to *cascading thresholds* [60], which is defined as the 'tendency of the crossing of one threshold to induce the crossing of other thresholds' [60]; here the changing human infection threshold also changes the sandfly population threshold with increasing treatment. The cascading thresholds are known to increase the resilience of one of the states [60]; here the resilience of the DFE is increased.

When more reservoirs, such as goats and cows, are available for sandflies to bite then scenarios for amplification (dilution) and subsequent increase (decrease) of the thresholds are possible; however, in all the possible scenarios, a shift in thresholds alters the region of stability of the endemic state. This also implies that multiple hosts either increase or decrease the resilience/persistence of DFE and endemic equilibria. For example, adding dead-end reservoirs to the BM, assuming equal biting preference for humans/reservoirs, lowers the equilibrium endemic state and in tandem the human infection threshold. Simultaneously, since some bites from sandflies are taken up by the dead-end hosts, a higher number of sandflies are required for the endemic state to emerge. Similarly, if the additional reservoirs are competitive for disease transmission, then a higher equilibrium endemic state–human infection prevalence threshold emerges, while requiring a lower number of sandflies. This follows from the fact that the competitive reservoirs act as a buffer for the disease, therefore, a smaller number of sandflies are required for endemic state to emerge. Additionally, the increasing and decreasing stability region of endemic state also follows if biting preferences are increased/decreased. Amplification/dilution of infection in different parameter regimes have also been studied in the context of host diversity although with density-dependent biting preference and fixed total population of all hosts [61]; this is the first time that an amplification and dilution phenomenon has been studied in the context of thresholds due to species diversity (here we assume population of individual reservoirs is a constant which is more realistic for visceral leishmaniasis).

Heterogeneity in sandfly biting behaviour implies that most bites come from a limited number of individuals. The interplay between sandfly population and host infection generates qualitative changes in respective thresholds: first, the force of infection is lowered, therefore, higher number of infectious hosts are required for sandflies to pick up sufficient infection, thereby increasing the infection invasion threshold for a given number of sandflies. Second, a new sandfly threshold is created which is higher than the corresponding homogeneous case. The higher sandfly threshold requires less infectious hosts for the endemic state to emerge.

The shifting thresholds indicate that control measures, such as treatment in humans and vector control used in combination with a control over reservoir population, easily stabilize the DFE. Thus, having multiple reservoirs can be beneficial if their population relative to humans can be managed given that the additional reservoirs are dead end. On the other hand, control will be much more difficult if the reservoirs are competitive and readily available for sandflies to bite.

Spatial heterogeneity and connectivity via human movement change thresholds in non-trivial ways. Our simple model shows that infection can persist for a lower sandfly population and human infection prevalence threshold as compared with the geographically isolated case due to weak migration. Thus, control efforts focused to bring down infection cases to less than 1 in 10 000 at the block level (in Bihar, India) will have to take into account of the fact that human movement lowers the local thresholds.

Sensitivity analysis using PRCC is able to identify parameters affecting the sandfly population threshold and human infection prevalence threshold. In particular, our results on PRCC show that sandfly threshold ($N_{Sc}$) decreases with an increase in biting rate and homogeneous mixing, the factor of asymptomatic/latent moving to infectious class, transmission parameters from sandfly to human (vice versa), natural mortality rate of humans, and proportion of infectious bites from PKDL cases ($\rho_2$). On the other hand, it is strongly but positively sensitive to sandfly mortality, human birth rate and disease-induced death rate. Similarly, the human infection prevalence threshold increases with the factor of asymptomatic/latent moving to infectious class, loss of immunity and heterogeneous biting pattern of sandflies, and it decreases with increasing disease-induced death rate, biting rate and treatment. Therefore, for control purposes it is desirable to reduce the contribution of PKDL cases by increasing the efficiency of treatment, decreasing the biting rates by covering the exposed skin and

reducing disease-induced deaths in addition to vector control. Our results also show that an optimal combination of treatment and vector control will need to be evaluated as the control programmes move forward.

This work highlights that despite control efforts, importation of cases can keep the VL transmission going due to feedback coming from continued movement from high endemic regions to low endemic regions. This is particularly relevant in areas like Muzaffarpur in Bihar India, where recently spatial patterns in VL transmission were studied [56,57]. Our results are based on simplistic assumptions on host competence and vector biting behaviour in the transmission cycle when multiple hosts are present. However, host competence is governed by the biology of the animals involved, while biting preference in general could be a nonlinear function of reservoir availability/population [44]. In these later circumstances analysing the equations with the corresponding functional dependencies will be complicated, we propose that the amplification and dilution behaviour presented in this work will be relevant. Consideration of a general vector-feeding-function based on host availability could be a project for future investigations. Although our two-patch model considered here captures the intuitive idea that lower thresholds can arise due to spatial heterogeneity, investigating a more realistic model by combining short-term [38] and long-term migration will be the goal for our future investigation.

Animal ethics. No animals or living things were involved for the study.

Research ethics. No approval research ethics approval was required.

Permission for fieldwork. No fieldwork was carried out.

Data accessibility. No data was used or generated during the study. Only numerical simulations were performed based on standard numerical algorithms, such as root finding using Newton–Raphson and bisection methods. Code is available at GitHub: https://github.com/shakir507/Lesihamaniasis and have been archived within the Zenodo repository: https://doi.org/10.5281/zenodo.4284376 [62]. The parameters used in the simulation are available in the papers cited.

Authors' contributions. Conceived and designed the experiments: E.M., A.M., S.B., R.C.R.; performed the experiments: S.B.; collected the data from published literature: S.B., R.C.R.; analysed the experimental results: S.B., R.C.R., E.M., A.M.; wrote the paper: S.B., E.M., A.M. All authors read and approved the final manuscript.

Competing interests. The authors declare no competing interests.

Funding. This work was made possible by the partial financial support provided by the National Institute of Health, USA on grant no. R01AI123245.

Acknowledgements. S.B. and E.M. acknowledge the financial support of the National Institute of Health, USA. S.B. also acknowledges SERB, Govt of India, for IndoUS postdoctoral fellowship.

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
