## [Reviewer comments · Royal Society Open Science]

Review History

RSOS-200904.R0 (Original submission)

Review form: Reviewer 1

Is the manuscript scientifically sound in its present form?

Yes

Are the interpretations and conclusions justified by the results?

Yes

Is the language acceptable?

Yes

Do you have any ethical concerns with this paper?

No

Have you any concerns about statistical analyses in this paper?

No

Recommendation?

Accept with minor revision (please list in comments)

Comments to the Author(s)

This manuscript analyzes a dynamical transmission model for visceral leishmaniasis (VL). I am a disease ecologist with expertise on vector-borne diseases and using dynamical models, but I am not a trained biomathematician. Based on my background, I could not find any errors in the analysis. However, I was unable to fully evaluate the mathematical model itself, so hopefully another reviewer had complimentary expertise to mine. I found the description of the model and results confusing in several places, and I have several suggestions to improve the presentation and explanation of the methods and results. My overall assessment is that, in its current state, this article is not adequately written to reach a broad audience of ecologists and epidemiologists interested in understanding the dynamics of visceral leishmaniasis transmission, which feels like a waste considering that the authors seem to want their results applied to VL control and the abstract and introduction felt like they were written for a broad audience (hence my accepting the invitation to review). With the revisions suggested below (I am calling them minor because they are regarding presentation rather than analysis), I think this manuscript could be a valuable contribution to the literature of vector-borne disease. As currently written, the target audience is more narrow, restricted to only people with training and interests specifically in theoretical or mathematical biology.

Major comments

Model description:

Page 4, line ~24: I would use the phrase “sandfly vectors” instead of “sandflies” when defining I_{vj} to make it easier for the reader to understand the meaning of the subscript v .

Equation 1: σ_j in the equations for dI_{1j}/dt and dR_{1j}/dt is missing a subscript; it should be σ_{1j} .

Equations 1-3: I strongly dislike the use of the Q_1 - Q_7 here. Please put the actual parameters in the equations. Having to flip around is confusing and requires more effort of the reader than using slightly longer equations. Table 1 is great, but it wouldn't take too much room to give basic definitions for the parameters here in the text (without specific subscripts, e.g., just say π_{ij} is birth rates, d_{ij} is disease induced death rates, etc.)

I also strongly dislike having the equation indices not match the equation labels, i.e., reservoir 1 gets subscript 2 and reservoir 2 gets subscript 3 (because humans got subscript 1). This problem could be easily fixed by simply calling them reservoir A and B instead.

Equation 5: Equations 1-4 are basic differential equations that do not need detailed explanations beyond defining the parameters. However, it is not obvious to me where the force of infection equations (eq. 5) come from or what they mean. Parameter N is never defined in the text or in table 1. I assume it's a generic way to describe counts for total humans or reservoirs, but that should be stated explicitly in the text. Parameters k and p are defined in the text, but not in Table 1.

A basic description of equation 5 to help the reader understand it would be nice. Is there an intuitive way to explain the different terms and how they relate to each other? There are two different summations in the equation – what do they each represent conceptually? It took me a long time to figure it out on my own; help make it easier for the average reader. (I believe) that the first summation is for multiple patches (maximum of 2 here, but could be more in theory), and the second one in the denominator is for the multiple hosts types (since there's transmission into each host type from all three host types). It's possible that all of that is immediately obvious to trained mathematicians, but I believe I am within the target audience for this article and it took me multiple passes to figure out what equation 5 was doing. Given that the journal does not have length limits, this is my biggest complaint, and what I think would improve the paper most. It might be useful to show multiple versions of the force of infection for the different models, if the

version without multiple hosts and patches is easier to understand and explain, and could help readers work their way to understanding the more complicated version. However, that might not be necessary if the complex version can be explained well enough.

I also don't understand the δ_{ji} notation at the bottom of equation 5. In the text it says δ_{ji} indicates non-human hosts don't move between patches, which makes sense as a concept, but I don't see how that matches the math. i and j are two different patches, so $i = j$ or $i \neq j$ seems like it would indicate the number of patches, not the movement or lack thereof of specific hosts between them (and there's nothing referring to subscripts for different hosts there).

Equation 7: I see now why the Q parameters might be useful here, for use in the various m equations. If you want to keep them, I would introduce them here and explicitly say that you are using them as a shorthand for the combined loss rate terms for the various host/vector categories.

Page 6, ~line 45: Best practice is to give the parameter (d_1) when talking about the disease induced death rate (as the authors do later in the text).

Page 8, line ~38: What does RHS mean?

Figures 2a, 2b, 2d, 3a, 5b, 7a, 7b, 7c, 7d: the x-axis on all of these panels is N_s , the "sandfly threshold", but in the text this parameter is referred to as N_{vc} (e.g., on page 8, lines ~43, 54, and 55) and v is the subscript used for all of the vector equations.

Figure 2: Is there a reason why panels a and b say "human prevalence" and panel d says "threshold prevalence"? Are these not the same y-axis? Figures 3 and 7 say "human prevalence" but Figure 5 says "threshold prevalence." Be consistent if they are indeed the same thing.

Improved explanations of key results:

Is there an intuitive, biological explanation for why disease-induced death causes the backward bifurcation? Disease-induced death removes infected humans from the system, which in theory should reduce transmission because they're no longer present to infect vectors. Is this what makes the DFE co-exist along with the EE, where otherwise only the EE would exist? Just a guess, but it would be nice if there was something like this to go along with the formal mathematical model.

Is there an intuitive, biological explanation for why the VL can persist deterministically if $R_0 < 1$? I browsed some of the relevant literature (e.g., Gumel 2012 JMAA) while reviewing this paper to help me understand this violation of what I had previously considered a tautology, and while I now understand that this is clearly a real mathematical phenomenon, but I don't understand biologically how it works for VL, or why disease-induced mortality induces it. Gumel 2012 presents a TB example where individuals can become reinfected from latent infections; in this case, disease maintenance when $R_0 < 1$ makes biological sense, because then the number of active infections doesn't depend solely on the horizontal transmission captured by R_0 . I did not understand the imperfect vaccine or vector-borne disease examples, since there was no biological explanation given. If R_0 is the average number of cases resulting from a single case, how can disease incidence increase if by definition each case causes fewer cases? There must be some influx of cases not captured by the equation for R_0 (as in the TB example, from latent infections), but where are they coming from in the case of VL? Again, I'm not a mathematician so perhaps I'm missing something obvious to the authors, but I think the paper's accessibility to vector-borne disease biologists such as myself would be increased by a more intuitive, biological explanation in addition to the formal mathematical analysis. On the one hand it feels unfair to demand an explanation like this, when the previously published literature on this topic does not adequately explain it in biological terms either, but I hope the authors can find a way to do it better.

Another perspective on this problem: in addition to being derived from mathematical models, R_0 (or R_{eff}) can also be estimated from epidemic time series or other empirical data (Heffernan et al 2005 Journal of the Royal Society Interface, Perspectives on the basic reproductive ratio, section 4), as so many people are now doing for COVID-19 (e.g., on the website [rti.live](https://www.rti.live)). If one were to try to estimate R_0 or R_{eff} from simulated time series of VL (or other vector-borne diseases) in the backwards bifurcation region where R_0 was < 1 according to the derived equation, would these “empirical” approaches yield estimates < 1 or > 1 ?

Discussion:

As written, the discussion just repeats the results section in slightly more detail without adding very much additional insight. The paper would be better if the discussion talked about how this study fits into the bigger context. Are backward bifurcations common in other vector-borne diseases? In other types of infectious diseases? Is there any evidence for backward bifurcations in any type of natural system? Is there evidence for disease amplification/dilution based on reservoir host competence and vector biting preference in other systems (e.g., zoonophylaxis with cows for malaria)? What does vector control for sandflies in India look like? Are there good data for sandfly population densities in India? What do we know about how the campaign to eradicate VL has actually been going? What are the biggest limitations of this study, both in terms of the assumptions it makes that might now be true, and in how it could be applied to real situations? What are the next steps for modeling? The authors don't necessarily need to address all of these questions, I'm just throwing out ideas for what might be interesting, but they should try to address some of them so the discussion doesn't function as just a results section 2.0, and helps readers put it in a larger context and better understand the significance of the study.

Minor comments

Introduction:

Page 2, line ~40: NTDs should be defined at first usage here, rather than below at second usage.

Page 2, line ~43: there is an “and” missing at the end of this list of mechanisms that can create multiple thresholds.

Page 2, line ~44: “vector-borne” typically has a dash

Page 2, line ~45: “frequency-dependent” typically has a dash.

Page 3, line ~14: This sentence here has a grammatical error: “although eliminate”

Page 3, line ~17: “it's” should not have an apostrophe

Page 3, line ~22: the aside “such as dogs (in Brazil for example)” should be set aside with a pair of commas to make the sentence easier to read

Page 3, line ~23: “however” should be preceded by a semi-colon instead of comma.

Page 3, line ~36: the aside “such as treatment” should be set aside with a pair of parentheses or commas to make the sentence easier to read

Page 3, line ~22: the aside “such as dogs (in Brazil for example)” should be set aside with a pair of commas to make the sentence easier to read

Page 3, line ~40-42: This section seems strange – most papers don't announce the flow of methods, results, and discussion, which is the standard structure. I can see why the more specific description of the methods section parts is useful, but it would be better incorporated into the previous paragraph.

Methods:

Page 4, line ~7: What does “in a native region” mean? A region with competent sandfly vectors? This is not a common phrase and should be replaced.

Page 4, line ~10: I would replace “VL transmission cycle” with either “the VL transmission cycle” or “VL transmission.” Also “human” should be “humans.”

Page 4, line ~15: would replace “specially” with “especially”

Page 4, line ~18-20: This sentence is a run-on sentence, split in two or re-write to be grammatically sound.

Page 4, line ~33: "loose" should be "lose"

Page 5, line ~28: "is indicates" should be "indicates." Also this is run-on sentence and should be split in two or re-written to be grammatically sound.

Page 5, line ~36-9: Also a run-on sentence that should be split in two or re-written to be grammatically sound. I think there's a "but" / "however" / "although" missing in the last half.

Page 6, line ~5: "the VL" should be "VL"

Page 6, line ~12: "state" should be "states"

Results:

Page 9, line ~23: The sentence starting with "although" is a fragment not a full sentence. Additionally, it's very strange and unorthodox to have Figure references sitting in the text as if they were complete sentences, rather than referred to parenthetically within a sentence (this also occurs below on line ~27).

Page 9, line ~28: semicolon before "however"

Page 9, line ~38: I would break this into three separate sentences to make it easier to read.

Discussion:

Page 10, line ~30: "threshold" should be "thresholds" and a colon or comma before "1)" would make it easier to read.

Page 10, line ~53: "a less number of sandflies" should be "a lower number of sandflies" or "fewer sandflies"

Page 11, line ~37: "host competence are" should be "host competence is" or "host competencies are"

Decision letter (RSOS-200904.R0)

Dear Dr Bilal

On behalf of the Editors, we are pleased to inform you that your Manuscript RSOS-200904 "Complexity and Critical Thresholds in the Dynamics of Visceral Leishmaniasis" has been accepted for publication in Royal Society Open Science subject to minor revision in accordance with the referees' reports. Please find the referees' comments along with any feedback from the Editors below my signature.

Please submit your revised manuscript and required files (see below) no later than 7 days from today's (ie 18-Sep-2020) date. Note: the ScholarOne system will 'lock' if submission of the revision is attempted 7 or more days after the deadline. If you do not think you will be able to meet this deadline please contact the editorial office immediately.

on behalf of Professor Tim Rogers (Associate Editor) and Pete Smith (Subject Editor)
openscience@royalsociety.org

Reviewer comments to Author:

Reviewer: 1

Comments to the Author(s)

This manuscript analyzes a dynamical transmission model for visceral leishmaniasis (VL). I am a disease ecologist with expertise on vector-borne diseases and using dynamical models, but I am not a trained biomathematician. Based on my background, I could not find any errors in the analysis. However, I was unable to fully evaluate the mathematical model itself, so hopefully another reviewer had complimentary expertise to mine. I found the description of the model and results confusing in several places, and I have several suggestions to improve the presentation and explanation of the methods and results. My overall assessment is that, in its current state, this article is not adequately written to reach a broad audience of ecologists and epidemiologists interested in understanding the dynamics of visceral leishmaniasis transmission, which feels like a waste considering that the authors seem to want their results applied to VL control and the abstract and introduction felt like they were written for a broad audience (hence my accepting the invitation to review). With the revisions suggested below (I am calling them minor because they are regarding presentation rather than analysis), I think this manuscript could be a valuable contribution to the literature of vector-borne disease. As currently written, the target audience is more narrow, restricted to only people with training and interests specifically in theoretical or mathematical biology.

Major comments

Model description:

Page 4, line ~24: I would use the phrase “sandfly vectors” instead of “sandflies” when defining I_{vj} to make it easier for the reader to understand the meaning of the subscript v .

Equation 1: σ_j in the equations for dI_{1j}/dt and dR_{1j}/dt is missing a subscript; it should be σ_{1j} .

Equations 1–3: I strongly dislike the use of the Q_1 – Q_7 here. Please put the actual parameters in the equations. Having to flip around is confusing and requires more effort of the reader than using slightly longer equations. Table 1 is great, but it wouldn’t take too much room to give basic definitions for the parameters here in the text (without specific subscripts, e.g., just say π_{ij} is birth rates, d_{ij} is disease induced death rates, etc.)

I also strongly dislike having the equation indices not match the equation labels, i.e., reservoir 1 gets subscript 2 and reservoir 2 gets subscript 3 (because humans got subscript 1). This problem could be easily fixed by simply calling them reservoir A and B instead.

Equation 5: Equations 1–4 are basic differential equations that do not need detailed explanations beyond defining the parameters. However, it is not obvious to me where the force of infection equations (eq. 5) come from or what they mean. Parameter N is never defined in the text or in table 1. I assume it's a generic way to describe counts for total humans or reservoirs, but that should be stated explicitly in the text. Parameters k and p are defined in the text, but not in Table 1.

A basic description of equation 5 to help the reader understand it would be nice. Is there an intuitive way to explain the different terms and how they relate to each other? There are two different summations in the equation – what do they each represent conceptually? It took me a long time to figure it out on my own; help make it easier for the average reader. (I believe) that the first summation is for multiple patches (maximum of 2 here, but could be more in theory), and the second one in the denominator is for the multiple hosts types (since there's transmission into each host type from all three host types). It's possible that all of that is immediately obvious to trained mathematicians, but I believe I am within the target audience for this article and it took me multiple passes to figure out what equation 5 was doing. Given that the journal does not have length limits, this is my biggest complaint, and what I think would improve the paper most. It might be useful to show multiple versions of the force of infection for the different models, if the version without multiple hosts and patches is easier to understand and explain, and could help readers work their way to understanding the more complicated version. However, that might not be necessary if the complex version can be explained well enough.

I also don't understand the δ_{ji} notation at the bottom of equation 5. In the text it says δ_{ji} indicates non-human hosts don't move between patches, which makes sense as a concept, but I don't see how that matches the math. i and j are two different patches, so $i = j$ or $i \neq j$ seems like it would indicate the number of patches, not the movement or lack thereof of specific hosts between them (and there's nothing referring to subscripts for different hosts there).

Equation 7: I see now why the Q parameters might be useful here, for use in the various m equations. If you want to keep them, I would introduce them here and explicitly say that you are using them as a shorthand for the combined loss rate terms for the various host/vector categories.

Page 6, ~line 45: Best practice is to give the parameter (d_1) when talking about the disease induced death rate (as the authors do later in the text).

Page 8, line ~38: What does RHS mean?

Figures 2a, 2b, 2d, 3a, 5b, 7a, 7b, 7c, 7d: the x-axis on all of these panels is N_s , the "sandfly threshold", but in the text this parameter is referred to as N_{vc} (e.g., on page 8, lines ~43, 54, and 55) and v is the subscript used for all of the vector equations.

Figure 2: Is there a reason why panels a and b say "human prevalence" and panel d says "threshold prevalence"? Are these not the same y-axis? Figures 3 and 7 say "human prevalence" but Figure 5 says "threshold prevalence." Be consistent if they are indeed the same thing.

Improved explanations of key results:

Is there an intuitive, biological explanation for why disease-induced death causes the backward bifurcation? Disease-induced death removes infected humans from the system, which in theory should reduce transmission because they're no longer present to infect vectors. Is this what

makes the DFE co-exist along with the EE, where otherwise only the EE would exist? Just a guess, but it would be nice if there was something like this to go along with the formal mathematical model.

Is there an intuitive, biological explanation for why the VL can persist deterministically if $R_0 < 1$? I browsed some of the relevant literature (e.g., Gumel 2012 JMAA) while reviewing this paper to help me understand this violation of what I had previously considered a tautology, and while I now understand that this is clearly a real mathematical phenomenon, but I don't understand biologically how it works for VL, or why disease-induced mortality induces it. Gumel 2012 presents a TB example where individuals can become reinfected from latent infections; in this case, disease maintenance when $R_0 < 1$ makes biological sense, because then the number of active infections doesn't depend solely on the horizontal transmission captured by R_0 . I did not understand the imperfect vaccine or vector-borne disease examples, since there was no biological explanation given. If R_0 is the average number of cases resulting from a single case, how can disease incidence increase if by definition each case causes fewer cases? There must be some influx of cases not captured by the equation for R_0 (as in the TB example, from latent infections), but where are they coming from in the case of VL? Again, I'm not a mathematician so perhaps I'm missing something obvious to the authors, but I think the paper's accessibility to vector-borne disease biologists such as myself would be increased by a more intuitive, biological explanation in addition to the formal mathematical analysis. On the one hand it feels unfair to demand an explanation like this, when the previously published literature on this topic does not adequately explain it in biological terms either, but I hope the authors can find a way to do it better.

Another perspective on this problem: in addition to being derived from mathematical models, R_0 (or Reffective) can also be estimated from epidemic time series or other empirical data (Heffernan et al 2005 Journal of the Royal Society Interface, Perspectives on the basic reproductive ratio, section 4), as so many people are now doing for COVID-19 (e.g., on the website [rt.live](https://www.rti.live)). If one were to try to estimate R_0 or Reff from simulated time series of VL (or other vector-borne diseases) in the backwards bifurcation region where $R_0 < 1$ according to the derived equation, would these "empirical" approaches yield estimates <1 or >1 ?

Discussion:

As written, the discussion just repeats the results section in slightly more detail without adding very much additional insight. The paper would be better if the discussion talked about how this study fits into the bigger context. Are backward bifurcations common in other vector-borne diseases? In other types of infectious diseases? Is there any evidence for backward bifurcations in any type of natural system? Is there evidence for disease amplification/dilution based on reservoir host competence and vector biting preference in other systems (e.g., zoonophylaxis with cows for malaria)? What does vector control for sandflies in India look like? Are there good data for sandfly population densities in India? What do we know about how the campaign to eradicate VL has actually been going? What are the biggest limitations of this study, both in terms of the assumptions it makes that might now be true, and in how it could be applied to real situations? What are the next steps for modeling? The authors don't necessarily need to address all of these questions, I'm just throwing out ideas for what might be interesting, but they should try to address some of them so the discussion doesn't function as just a results section 2.0, and helps readers put it in a larger context and better understand the significance of the study.

Minor comments

Introduction:

Page 2, line ~40: NTDs should be defined at first usage here, rather than below at second usage.
Page 2, line ~43: there is an "and" missing at the end of this list of mechanisms that can create multiple thresholds.

Page 2, line ~44: "vector-borne" typically has a dash

Page 2, line ~45: "frequency-dependent" typically has a dash.

Page 3, line ~14: This sentence here has a grammatical error: "although eliminate"

Page 3, line ~17: "it's" should not have an apostrophe

Page 3, line ~22: the aside "such as dogs (in Brazil for example)" should be set aside with a pair of commas to make the sentence easier to read

Page 3, line ~23: "however" should be preceded by a semi-colon instead of comma.

Page 3, line ~36: the aside "such as treatment" should be set aside with a pair of parentheses or commas to make the sentence easier to read

Page 3, line ~22: the aside "such as dogs (in Brazil for example)" should be set aside with a pair of commas to make the sentence easier to read

Page 3, line ~40-42: This section seems strange - most papers don't announce the flow of methods, results, and discussion, which is the standard structure. I can see why the more specific description of the methods section parts is useful, but it would be better incorporated into the previous paragraph.

Methods:

Page 4, line ~7: What does "in a native region" mean? A region with competent sandfly vectors? This is not a common phrase and should be replaced.

Page 4, line ~10: I would replace "VL transmission cycle" with either "the VL transmission cycle" or "VL transmission." Also "human" should be "humans."

Page 4, line ~15: would replace "specially" with "especially"

Page 4, line ~18-20: This sentence is a run-on sentence, split in two or re-write to be grammatically sound.

Page 4, line ~33: "loose" should be "lose"

Page 5, line ~28: "is indicates" should be "indicates." Also this is run-on sentence and should be split in two or re-written to be grammatically sound.

Page 5, line ~36-9: Also a run-on sentence that should be split in two or re-written to be grammatically sound. I think there's a "but" / "however" / "although" missing in the last half.

Page 6, line ~5: "the VL" should be "VL"

Page 6, line ~12: "state" should be "states"

Results:

Page 9, line ~23: The sentence starting with "although" is a fragment not a full sentence. Additionally, it's very strange and unorthodox to have Figure references sitting in the text as if they were complete sentences, rather than referred to parenthetically within a sentence (this also occurs below on line ~27).

Page 9, line ~28: semicolon before "however"

Page 9, line ~38: I would break this into three separate sentences to make it easier to read.

Discussion:

Page 10, line ~30: "threshold" should be "thresholds" and a colon or comma before "1)" would make it easier to read.

Page 10, line ~53: "a less number of sandflies" should be "a lower number of sandflies" or "fewer sandflies"

Page 11, line ~37: "host competence are" should be "host competence is" or "host competencies are"

===PREPARING YOUR MANUSCRIPT===

- one version identifying all the changes that have been

made (for instance, in coloured highlight, in bold text, or tracked changes);a 'clean' version of the new manuscript that incorporates the changes made, but does not highlight them. This version will be used for typesetting.
Please ensure that any equations included in the paper are editable text and not embedded images.

===PREPARING YOUR REVISION IN SCHOLARONE===

- If you are providing image files for potential cover images, please upload these at this step, and inform the editorial office you have done so. You must hold the copyright to any image provided.
- A copy of your point-by-point response to referees and Editors. This will expedite the preparation of your proof.

- Ensure that your data access statement meets the requirements at <https://royalsociety.org/journals/authors/author-guidelines/#data>. You should ensure that you cite the dataset in your reference list. If you have deposited data etc in the Dryad repository, please only include the 'For publication' link at this stage. You should remove the 'For review' link.
- If you are requesting an article processing charge waiver, you must select the relevant waiver option (if requesting a discretionary waiver, the form should have been uploaded at Step 3 'File upload' above).
- If you have uploaded ESM files, please ensure you follow the guidance at <https://royalsociety.org/journals/authors/author-guidelines/#supplementary-material> to include a suitable title and informative caption. An example of appropriate titling and captioning may be found at [https://figshare.com/articles/Table_S2_from_Is_there_a_trade-off_between_peak_performance_and_performance_breadth_across_temperatures_for_aerobic_sc ope_in_teleost_fishes_/3843624](https://figshare.com/articles/Table_S2_from_Is_there_a_trade-off_between_peak_performance_and_performance_breadth_across_temperatures_for_aerobic_scope_in_teleost_fishes_/3843624).

Author's Response to Decision Letter for (RSOS-200904.R0)

See Appendix A.

Decision letter (RSOS-200904.R1)

Dear Dr Bilal,

It is a pleasure to accept your manuscript entitled "Complexity and Critical Thresholds in the Dynamics of Visceral Leishmaniasis" in its current form for publication in Royal Society Open Science.

You can expect to receive a proof of your article in the near future. Please contact the editorial office (openscience_proofs@royalsociety.org) and the production office

(openscience@royalsociety.org) to let us know if you are likely to be away from e-mail contact -- if you are going to be away, please nominate a co-author (if available) to manage the proofing process, and ensure they are copied into your email to the journal.

on behalf of Professor Tim Rogers (Associate Editor) and Pete Smith (Subject Editor)
openscience@royalsociety.org

Appendix A

Reviewer comments to Author:

Reviewer: 1

Comments to the Author(s):

This manuscript analyzes a dynamical transmission model for visceral leishmaniasis (VL). I am a disease ecologist with expertise on vector-borne diseases and using dynamical models, but I am not a trained biomathematician. Based on my background, I could not find any errors in the analysis. However, I was unable to fully evaluate the mathematical model itself, so hopefully another reviewer had complimentary expertise to mine. I found the description of the model and results confusing in several places, and I have several suggestions to improve the presentation and explanation of the methods and results. My overall assessment is that, in its current state, this article is not adequately written to reach a broad audience of ecologists and epidemiologists interested in understanding the dynamics of visceral leishmaniasis transmission, which feels like a waste considering that the authors seem to want their results applied to VL control and the abstract and introduction felt like they were written for a broad audience (hence my accepting the invitation to review). With the revisions suggested below (I am calling them minor because they are regarding presentation rather than analysis), I think this manuscript could be a valuable contribution to the literature of vector-borne disease. As currently written, the target audience is more narrow, restricted to only people with training and interests specifically in theoretical or mathematical biology.

Response: We thank the reviewer for agreeing that our article and work would make a valuable contribution to modelling vector-borne diseases. We understand the concerns regarding broader engagement and have paid careful attention to the reviewer's suggestions on how to do this better in this article.

Major comments:

Model description:

Comment: Page 4, line ~24: I would use the phrase "sandfly vectors" instead of "sandflies" when defining I_{vj} to make it easier for the reader to understand the meaning of the subscript v .

Response: Thank you for suggesting this phrase. We have now incorporated it.

Comment:

Equation 1: σ_j in the equations for dI_{1j}/dt and dR_{1j}/dt is missing a subscript; it should be σ_{1j} .

Response: We have now corrected it now. Additionally, we have now used a more consistent subscripts highlighting human, non-human reservoirs, and sandfly vector related parameters and compartments in the model.

Comment:

Equations 1–3: I strongly dislike the use of the Q_1 – Q_7 here. Please put the actual parameters in the equations. Having to flip around is confusing and requires more effort of the reader than using slightly longer equations. Table 1 is great, but it wouldn't take too much room to give basic definitions for the parameters here in the text (without specific subscripts, e.g., just say π_{ij} is birth rates, d_{ij} is disease induced death rates, etc.)

I also strongly dislike having the equation indices not match the equation labels, i.e., reservoir 1 gets subscript 2 and reservoir 2 gets subscript 3 (because humans got subscript 1). This problem could be easily fixed by simply calling them reservoir A and B instead.

Response: We have now rewritten this part of the paper and incorporated suggested changes. Thank you.

Comment:

Equation 5: Equations 1–4 are basic differential equations that do not need detailed explanations beyond defining the parameters. However, it is not obvious to me where the force of infection equations (eq. 5) come from or what they mean. Parameter N is never defined in the text or in table 1. I assume it's a generic way to describe counts for total humans or reservoirs, but that should be stated explicitly in the text. Parameters k and p are defined in the text, but not in Table 1.

A basic description of equation 5 to help the reader understand it would be nice. Is there an intuitive way to explain the different terms and how they relate to each other? There are two different summations in the equation – what do they each represent conceptually? It took me a long time to figure it out on my own; help make it easier for the average reader. (I believe) that the first summation is for multiple patches (maximum of 2 here, but could be more in theory), and the second one in the denominator is for the multiple hosts types (since there's transmission into each host type from all three host types). It's possible that all of that is immediately obvious to trained mathematicians, but I believe I am within the target audience for this article and it took me multiple passes to figure out what equation 5 was doing. Given that the journal does not have length limits, this is my biggest complaint, and what I think would improve the paper most. It might be useful to show multiple versions of the force of infection for the different models, if the version without multiple hosts and patches is easier to understand and explain, and could help readers work their way to understanding the more complicated version. However, that might not be necessary if the complex version can be explained well enough.

I also don't understand the Δ_{ij} notation at the bottom of equation 5. In the text it says Δ_{ij} indicates non-

human hosts don't move between patches, which makes sense as a concept, but I don't see how that that matches the math. i and j are two different patches, so $i = j$ or $i \neq j$ seems like it would indicate the number of patches, not the movement or lack thereof of specific hosts between them (and there's nothing referring to subscripts for different hosts there).

Response: Parameter N , k , p are now defined both in the text as well as in table 1. The force of infection builds upon previous literature only to be extended for a multihost model like we present in the paper. We have now included a description (in the text and Supplementary Document) of how the current force of infection can be derived following previously published work. Since we have now described the parameters in the text and also in the table, the confusion regarding Δ_{ij} is resolved, we acknowledge though in the previous state it was confusing.

Comment: Equation 7: I see now why the Q parameters might be useful here, for use in the various m equations. If you want to keep them, I would introduce them here and explicitly say that you are using them as a shorthand for the combined loss rate terms for the various host/vector categories.

Response: We agree with the suggestion made and have incorporated it now. Thank you.

Comment: Page 6, ~line 45: Best practice is to give the parameter (d_1) when talking about the disease induced death rate (as the authors do later in the text).

Response: Included now. Should have been there in the first instance. Thank you for pointing this out.

Comment: Page 8, line ~38: What does RHS mean?

Response: It's an abbreviation for right hand side, we have replaced it with the full phrase.

Comment: Figures 2a, 2b, 2d, 3a, 5b, 7a, 7b, 7c, 7d: the x-axis on all of these panels is N_s , the "sandfly threshold", but in the text this parameter is referred to as N_{vc} (e.g., on page 8, lines ~43, 54, and 55) and v is the subscript used for all of the vector equations.

Response: N_s is sandfly population parameter, where as N_{vc} is its threshold value below which only disease free state exists. We agree that we can make N_{vc} as N_{sc} to indicate threshold value of sandfly population. And that we have made these changes now in the manuscript.

Comment: Figure 2: Is there a reason why panels a and b say “human prevalence” and panel d says “threshold prevalence”? Are these not the same y-axis? Figures 3 and 7 say “human prevalence” but Figure 5 says “threshold prevalence.” Be consistent if they are indeed the same thing.

Response: We are calling the threshold value of human infection prevalence as the threshold prevalence. This is done with the understanding that we are talking of only human infection prevalence and only look at it’s threshold value, wherever threshold prevalence is mentioned. One can notice that human prevalence is mentioned in bifurcation diagrams whereas threshold in fig 2(d) and Fig5 are threshold values of human infection prevalence at the sandfly population threshold. We believe it is consistent with what we are saying. We have now referred to threshold human prevalence in the context of figure 5 in the text.

Comment: Improved explanations of key results-Is there an intuitive, biological explanation for why disease-induced death causes the backward bifurcation? Disease-induced death removes infected humans from the system, which in theory should reduce transmission because they’re no longer present to infect vectors. Is this what makes the DFE co-exist along with the EE, where otherwise only the EE would exist? Just a guess, but it would be nice if there was something like this to go along with the formal mathematical model.

Is there an intuitive, biological explanation for why the VL can persist deterministically if $R_0 < 1$? I browsed some of the relevant literature (e.g., Gumel 2012 JMAA) while reviewing this paper to help me understand this violation of what I had previously considered a tautology, and while I now understand that this is clearly a real mathematical phenomenon, but I don’t understand biologically how it works for VL, or why disease-induced mortality induces it. Gumel 2012 presents a TB example where individuals can become reinfected from latent infections; in this case, disease maintenance when $R_0 < 1$ makes biological sense, because then the number of active infections doesn’t depend solely on the horizontal transmission captured by R_0 . I did not understand the imperfect vaccine or vector-borne disease examples, since there was no biological explanation given. If R_0 is the average number of cases resulting from a single case, how can disease incidence increase if by definition each cases causes fewer cases? There must be some influx of cases not captured by the equation for R_0 (as in the TB example, from latent infections), but where are they coming from in the case of VL? Again, I’m not a mathematician so perhaps I’m missing something obvious to the authors, but I think the paper’s accessibility to vector-borne disease biologists such as myself would be increased by a more intuitive, biological explanation in addition to the formal mathematical analysis. On the one hand it feels unfair to demand an explanation like this, when the previously published literature on this topic does not adequately explain it in biological terms either, but I hope the authors can find a way to do it better.

Response: We understand the reviewers concern and we think the following point might address some of your concerns.

- Similar to gain in I_h compartment due to reinfection in TB model, increments in infection occur in our model due to increments in I_v (infected number of vectors) via conditions of backward

bifurcation. In addition to disease death parameter, the parameters that increases I_v are important even for existence of the backward bifurcation in our model. We have now clarified clearly the role of other parameters in backward bifurcation in the main text of the manuscript (see Discussion section and also Supplementary Document).

- Note, by definition, R_0 is quantified at the *beginning of the local outbreak*, when most individuals are susceptible. On the other hand, backward bifurcation is observed due to three factors, an increase in diseased induced death rate above its critical value (which brings R_0 below one), an initial (at time zero) size of infection (I_0), and death rate and infection rate of host, which have impact on I_v . If this initial size of infection is significantly higher than its threshold value than disease might become endemic even when $R_0 < 1$.
- An example of this backward bifurcation could be, chikungunya in South America, which was not endemic before 2000's even though a regular injection of chikungunya cases into the South America from various parts of the affected world, via air-travel. Regular outbreaks of chikungunya are now observed in the South America. We have now introduced an argument to this effect in the discussion section.
- The conditions of backward bifurcation are function of three parameters: disease deaths, infection rate of host on vector and average life span of host. If backward bifurcation exists in the system, then there are two parameters that decide the eventual dynamics of the system: R_0 (which should be less than one) and initial size of infection (I_{h_0} and I_{v_0}).

Comment: Another perspective on this problem: in addition to being derived from mathematical models, R_0 (or Reffective) can also be estimated from epidemic time series or other empirical data (Heffernan et al 2005 Journal of the Royal Society Interface, Perspectives on the basic reproductive ratio, section 4), as so many people are now doing for COVID-19 (e.g., on the website [rt.live](https://www.rtt.live/)). If one were to try to estimate R_0 or R_{eff} from simulated time series of VL (or other vector-borne diseases) in the backwards bifurcation region where $R_0 < 1$ according to the derived equation, would these "empirical" approaches yield estimates < 1 or > 1 ?

Response: Thank you for an interesting observation and your query.

- Heffernan et al. 2005 provides 4 different approaches to estimate R_0 using epidemic time series or prevalence. Note, all these 4 approaches are derived from a mathematical model but under different assumption. For example, R_0 can be estimated from the initial exponential rise in incidence or from prevalence and equilibrium expression if disease is endemic. In all these estimation methods, most parameters were kept fixed from the literature during the estimation procedure to estimate one or two parameters of the model.
- $R_{eff}(t)$ is usually computed using different methods as it needs the quantity (number of susceptibles) that change over time. Sometimes branching processes can be used after the initial exponential phase (once interventions have been initiated) to estimate $R_{eff}(t)$.
- Although, we do not attempt to provide estimates of R_0 from our theoretical study, but we think endemic prevalence and equilibrium expression can be used to estimate R_0 for VL.

Comment: Discussion-As written, the discussion just repeats the results section in slightly more detail without adding very much additional insight. The paper would be better if the discussion talked about how this study fits into the bigger context. Are backward bifurcations common in other vector-borne diseases? In other types of infectious diseases? Is there any evidence for backward bifurcations in any type of natural system? Is there evidence for disease amplification/dilution based on reservoir host competence and vector biting preference in other systems (e.g., zoonophylaxis with cows for malaria)? What does vector control for sandflies in India look like? Are there good data for sandfly population densities in India? What do we know about how the campaign to eradicate VL has actually been going? What are the biggest limitations of this study, both in terms of the assumptions it makes that might now be true, and in how it could be applied to real situations? What are the next steps for modeling? The authors don't necessarily need to address all of these questions, I'm just throwing out ideas for what might be interesting, but they should try to address some of them so the discussion doesn't function as just a results section 2.0, and helps readers put it in a larger context and better understand the significance of the study.

Response: Thank you for your comments. We now provide more details on backward bifurcation and its examples in the discussion section. We also try to provide answers to most of your questions raised.

Backward bifurcations belong to a class of disease systems wherein addition to the basic reproduction number, initial infection levels act as a threshold for deciding eventual trends from the system. Such systems have been observed in both micro- and macro-parasitic directly-transmitted as well as vector-transmitted diseases, such as, TB, dengue Lymphatic filariasis, and Onchocerciasis. Although the microparasitic systems have been explored both in the field as well in theoretical literature, the one theme which it has consistent is that control is not trivial in systems exhibiting multiple thresholds.

This provides the motivation to investigate situations if a microparasitic vector-borne disease like VL might exhibit multiple thresholds, and what the implications would be vis-a-vis control when additional ecological complexity of multiple hosts, biting heterogeneity and spatial heterogeneity is accounted for.

The initial infection prevalence has been recently understood to be a driving factor behind emergence of chikungunya in south America, where despite international travel the disease could not become endemic until a threshold of imported infections was reached. Importantly the imported infection could not be accounted for in calculations of R_0 which would have been less than one because the disease was not endemic in the first place.

We have now highlighted two things related to a vector borne disease: parameters of the model that are critical for existence of backward bifurcation (disease deaths, natural death rates in host, and transmission rates of host on vector) and quantities that are needed for eventual trend of system having backward bifurcation (R_0 and the initial number of infections in both host and vectors).

Minor comments

Introduction:

Comment:

Page 2, line ~40: NTDs should be defined at first usage here, rather than below at second usage.

Page 2, line ~43: there is an “and” missing at the end of this list of mechanisms that can create multiple thresholds.

Page 2, line ~44: “vector-borne” typically has a dash

Page 2, line ~45: “frequency-dependent” typically has a dash.

Page 3, line ~14: This sentence here has a grammatical error: “although eliminate”

Page 3, line ~17: “it’s” should not have an apostrophe

Page 3, line ~22: the aside “such as dogs (in Brazil for example)” should be set aside with a pair of commas to make the sentence easier to read

Page 3, line ~23: “however” should be preceded by a semi-colon instead of comma.

Page 3, line ~36: the aside “such as treatment” should be set aside with a pair of parentheses or commas to make the sentence easier to read

Page 3, line ~22: the aside “such as dogs (in Brazil for example)” should be set aside with a pair of commas to make the sentence easier to read

Response: Incorporated these suggestions, beginning from page 2 line 40 to page 3 line 22 covering the introduction section. Thank you.

Comment: Page 3, line ~40-42: This section seems strange – most papers don’t announce the flow of methods, results, and discussion, which is the standard structure. I can see why the more specific description of the methods section parts is useful, but it would be better incorporated into the previous paragraph.

Response: We see where reviewer is coming from. We have incorporated the suggestion.

Comment:

Methods:

Page 4, line ~7: What does “in a native region” mean? A region with competent sandfly vectors? This is not a common phrase and should be replaced.

Page 4, line ~10: I would replace “VL transmission cycle” with either “the VL transmission cycle” or “VL transmission.” Also “human” should be “humans.”

Page 4, line ~15: would replace “specially” with “especially”

Page 4, line ~18-20: This sentence is a run-on sentence, split in two or re-write to be grammatically sound.

Page 4, line ~33: “loose” should be “lose”

Page 5, line ~28: “is indicates” should be “indicates.” Also this is run-on sentence and should be split in

two or re-written to be grammatically sound.

Page 5, line ~36-9: Also a run-on sentence that should be split in two or re-written to be grammatically sound. I think there's a "but" / "however" / "although" missing in the last half.

Page 6, line ~5: "the VL" should be "VL"

Page 6, line ~12: "state" should be "states"

Response: We have incorporated all the changes suggested for the methods section. Thank you for pointing them out in such detail.

Comment:

Results:

Page 9, line ~23: The sentence starting with "although" is a fragment not a full sentence. Additionally, it's very strange and unorthodox to have Figure references sitting in the text as if they were complete sentences, rather than referred to parenthetically within a sentence (this also occurs below on line ~27).

Page 9, line ~28: semicolon before "however"

Page 9, line ~38: I would break this into three separate sentences to make it easier to read.

Response: We have addressed these concerns for the results section. Thank you for pointing them out in such detail.

Comment:

Discussion:

Page 10, line ~30: "threshold" should be "thresholds" and a colon or comma before "1)" would make it easier to read.

Page 10, line ~53: "a less number of sandflies" should be "a lower number of sandflies" or "fewer sandflies"

Page 11, line ~37: "host competence are" should be "host competence is" or "host competencies are"

Response: We have addressed these concerns for the discussion section. Thank you for pointing them out in such detail.